# Studying volatility from composition, dilution, and heating measurements of secondary organic aerosols formed during α-pinene ozonolysis

Kei Sato[1], Yuji Fujitani[1], Satoshi Inomata[1], Yu Morino[1], Kiyoshi Tanabe[1], Sathiyamurthi Ramasamy[1], Toshihide Hikida[2], Akio Shimono,[2] Akinori Takami[1], Akihiro Fushimi[1], Yoshinori Kondo[1], Takashi Imamura[1], Hiroshi Tanimoto[1], Seiji Sugata[1]

[1]National Institute for Environmental Studies, Ibaraki, 305-8506, Japan
[2]Shoreline Science Research, Inc., Tokyo, 192-0045, Japan

*Correspondence to*: Kei Sato (kei@nies.go.jp)

**Abstract.** Traditional yield curve analysis shows that semi-volatile organic compounds are a major component of secondary organic aerosols (SOAs). We investigated the volatility distribution of SOAs from α-pinene ozonolysis using positive electrospray ionization mass analysis and dilution- and heat-induced evaporation measurements. Laboratory chamber experiments were conducted on α-pinene ozonolysis, in the presence and absence of OH scavengers. Among these, we identified not only semi-volatile products, but also less volatile highly oxygenated molecules (HOMs) and dimers. Ozonolysis products were further exposed to OH radicals to check the effects of photochemical aging. HOMs were also formed during OH-initiated photochemical aging. Most HOMs that formed from ozonolysis and photochemical aging had ten or less carbons. SOA particle evaporation after instantaneous dilution was measured at <1% and ~40% relative humidity. The volume fraction remaining of SOAs decreased with time and the equilibration time scale was determined to be 24 – 46 min for SOA evaporation. The experimental results of equilibration time scale could be explained when the mass accommodation coefficient is assumed to be 0.1, suggesting that the existence of low-volatility materials in SOAs, kinetic inhibition, or some combined effect may explain the equilibration time scale measured in this study.

## 1    Introduction

Atmospheric fine aerosols are believed to affect climate (IPCC, 2013) and human health (Dochery et al., 1993). Secondary organic aerosols (SOAs) are a major component of atmospheric fine aerosols (Zhang et al., 2007). Volatility basis-set (VBS) models have improved the prediction of atmospheric SOA levels by accounting for the decrease in SOA volatility with photochemical aging (Robinson et al., 2007). Our group recently compared the results of VBS model simulations with observational data from several urban and rural sites leeward of mainland East Asia (Morino et al., 2015). The VBS model improved the prediction of ambient organic aerosol levels in spring and summer, but still included large uncertainties.

The volatility distribution of SOAs, a key property in the prediction of particle levels, has been investigated by various techniques. The volatility distribution was evaluated from the chamber results of a SOA yield curve (Lane et al., 2008). This analysis assumes single-generation oxidation and instantaneous gas-particle partitioning for chamber experiments. Another technique to study particle volatility distribution is heat-induced evaporation. This technique was often applied for SOAs formed from the ozonolysis of α-pinene, a typical SOA source in the troposphere (Huffman et al, 2009; Epstein et al., 2010; Salo et al., 2011; Kolesar et al., 2015; Saha and Grieshop, 2016). Heat-induced evaporation provides the volatility at high temperatures, and the enthalpy of vaporization is

needed to determine the volatility at ambient temperatures. Furthermore, thermal decomposition may affect results

obtained by this method.

Yet another technique used to study volatility distribution is dilution-induced evaporation, which has been successfully applied to the volatility studies of diesel exhaust particles (Robinson et al., 2007; Fujitani et al., 2012). Grieshop et al. (2007) diluted SOA particles in a reactor and studied the reversibility of gas–particle partitioning. Later workers (Vaden et al., 2011; Saleh et al.; 2013; Wilson et al., 2015; Yli-Juuti et al., 2017)

diluted SOA particles instantaneously in an external chamber. Saleh et al. (2011; 2013) defined the equilibration time scale of SOA evaporation, and reported that the equilibration time scale is several minutes to several tens of minutes for α-pinene SOA particles (Saleh et al., 2013). Slow evaporation could be due to the presence of low-volatility materials in SOAs, kinetic inhibition, or some combined effect (Saha and Grieshop, 2016). Therefore, data from dilution-induced evaporation measurements would be determined not only by product

volatility but also by the particle phase.

The chemical analysis of particles can also provide information on the volatility and formation mechanisms of products. Chemical or electrospray ionization mass spectrometry can identify a wide range of oxygenated organic molecules. Using these techniques revealed that the major particle products from α-pinene ozonolysis are highly oxygenated molecules (HOMs), ester dimers, and semi-volatile compounds (Zhang et al., 2015; 2017). Recent

studies on the parameterization of saturation concentration (Shiraiwa et al., 2014; Li et al., 2016) and the sensitivity of electrospray ionization mass spectrometry (Kruve et al., 2013; Heinritzi et al., 2016) is helpful to evaluate SOA volatility using mass analysis data.

HOMs are believed to be multifunctional peroxides (Ehn et al., 2014) or multifunctional acids (Szmigielski et al., 2007). HOMs would be formed through the auto-oxidation of organic peroxy ($RO_2$) radicals (Zhang et al., 2017) or

organic oxy (RO) radicals (Müller et al., 2012). Ester dimers identified in α-pinene SOAs might be produced by the particle-phase dehydration (Yasmeen et al., 2010), reactions of stabilised Criegee intermediates with other organic molecules (Kristensen et al., 2014; 2016), or gas reactions between two acylperoxy radicals followed by heterogeneous processes (Zhang et al., 2015).

In this study, we performed chamber experiments on α-pinene ozonolysis in the presence and absence of OH

scavengers. Ozonolysis products were further exposed to OH radicals to check the effect of photochemical aging. The volatility distribution of SOAs were evaluated with a wide range of techniques, including offline chemical analysis and dilution- and heat-induced evaporation. We employed positive electrospray ionization mass analysis to detect HOMs and dimers (Zhang et al., 2017). This work aims to evaluate the volatility distribution of α-pinene SOAs using three different techniques and discuss SOA formation processes for improvement of current

atmospheric models.

## 2    Methods

### 2.1. Chamber experiment

Figure S1 shows a schematic diagram of the laboratory chamber system and analytical instruments used in this study. Initial concentrations are listed with the mass concentration and mean size of the SOA particles produced

(Table 1). A 6 m$^3$ Teflon-coated stainless steel chamber (Sato et al., 2007) was used for experiments under dry conditions (runs 1–9). Prior to each experiment, the chamber was filled with purified air from a custom-made air purifier (Horiba Stec Ltd., Japan; relative humidity < 1%). The temperature of the chamber was controlled at $298 \pm 1$ K. Required amounts of α-pinene (0.15–0.84 ppmv) and ozone (0.53–1.13 ppmv) were injected into the chamber. In runs 1–4, diethyl ether was also added as an OH scavenger, to suppress secondary reactions with OH radicals

from the ozonolysis. In run 6, CO was used as an alternative OH scavenger. In run 9, 1 ppmv methyl nitrite was
     added as an OH source after α-pinene was consumed, and the mixture was then irradiated by light from 19 xenon
     lamps (1 kW each); the light was passed through Pyrex filters. The $NO_2$ photolysis rate was 0.29 $min^{-1}$. The
     concentrations of α-pinene, ozone, pinonaldehyde (PA), and methyl nitrite were measured every 6 min with an
     *in-situ* Fourier transform-infrared (FT-IR) spectrometer (Nexus 670, Thermo Fisher Scientific, United States of
America (USA)) with a 221.5 m optical path. Experiments under humid conditions (runs 10–12) were performed
     with a 6 $m^3$ fluorinated ethylene polyethylene (FEP) film bag (1.81 × 1.81 × 1.81 m, 50 μm). The clean air was
     supplied by purified air generator (DAR-2200, Horiba Ltd., Japan). The relative humidity (RH) measured in the
     bag was ~40%. The temperature of the laboratory was controlled at 298 ± 1 K. The ozone concentration in the bag
     was monitored every 1 min by an ozone monitor (Model 1150, Dylec Inc., Japan). For dilution-induced
evaporation measurements, ~20 ppmv CO was added to the reaction mixture as a dilution marker. The CO levels
     before and after dilution were measured using a CO monitor (Model 48i-TLE, Thermo Fisher Scientific, USA).
     The coefficient of determination ($R^2$) was > 0.99 for the linear least squares analysis of the CO calibration data (3.5
     –168 ppmv, n = 8).

     Particle size distribution was observed every 3 min using a scanning mobility particle sizer (SMPS) (Model 3034,
TSI Inc., USA). The effective density of the particles was measured using a custom-made differential mobility
     analyser (DMA) (Sibata Scientific Technology Ltd., Japan), an aerosol particle mass analyser (APM) (Model 3600,
     Kanomax Inc., Japan), and a condensation particle counter (CPC) (Model 3772, TSI Inc., USA). The effective
     density of α-pinene ozonolysis SOA was determined to be 1.34 ± 0.12 g $cm^{-3}$. The density determined in this study
     is close to literature values: 1.32 ± 0.10 g $cm^{-3}$ (Ng et al. 2007) and 1.24 ± 0.03 (Malloy et al., 2009).
A high-resolution time-of-flight aerosol mass spectrometer (AMS) (H-ToF-AMS, Aerodyne Research, USA)
     (Aiken et al., 2008) combined with a thermodenuder (TD) (ARI thermal denuder, Aerodyne Research, USA)
     (Faulhaber et al., 2009) was used to measure heat-induced evaporation. The residence time in the heating zone of
     thermodenuder was ~13 s at room temperature. SOA mass concentration was measured every 3 min using the
     AMS. The TD was equipped with a bypass line. During each cycle of TD-AMS measurement, we used the
bypass for first 9 min to obtain the reference data and then used the TD for 15 min to obtain the data of a specific
     temperature. Pinonic acid particles generated from an aqueous solution of pinonic acid (13 mM) using an aerosol
     generator (Model 3076, TSI Inc., USA), and then dried with a diffusion dryer (Model 3062, TSI Inc., USA), were
     used as a reference for the TD-AMS measurements.

     Gas and particle products were measured using a proton transfer reaction-mass spectrometer (PTR-MS)
(PTR-QMS 500, Ionicon Analytik, Austria) (Lindinger et al., 1998) to determine the saturation concentration of
     each product. The detailed procedure has been explained elsewhere (Inomata et al., 2014). To measure the
     products in the gas phase, online measurements were taken from the filtered chamber air after the reaction finished.
     After this, products in the particle phase were measured; particles were collected on a Fluoropore Teflon filter
     (Sumitomo Electric Industries, Japan; 47 mmφ, pore size 1 μm) at 16.7 L $min^{-1}$ for 30 min. The sample filter was
placed in a filter holder which was then heated at 368 K under a stream of nitrogen; the gases that evaporated from
     the filter were measured using the PTR-MS. The saturation concentration was calculated from gas-particle ratio
     measured by the PTR-MS for each *m/z*, assuming gas-particle equilibrium. The ion signals of m/z 151–229 were
     only used for the evaluations of saturation concentration. The signals of *m/z* < 151 were not used because there
     would be interference from fragment ions. The signals of *m/z* > 229 were not detected due to the low sensitivity
of the quadruple mass spectrometer.

### 2.2. LC/MS analysis

Offline chemical analysis was performed using the procedure described in Sato et al. (2016). We used positive electrospray ionization-liquid chromatograph/time-of-flight-mass spectrometry (LC/MS). After the reaction finished, particle products were collected on another Teflon filter (Sumitomo Electric Industries, Japan, 47 mmφ, pore size 1 μm) at 16.7 L min⁻¹ for 30 min. After sampling, the filter sample was sonicated in 5 mL methanol for 30 min using an ultrasonic bath (Model UT-105S, Sharp, Japan; 130 W). The filter extract was concentrated to near dryness under a gentle stream of nitrogen (typically ~1 L min⁻¹). A 1 mL formic acid–methanol–water solution (v/v/v = 0.05/100/99.95) was added to the concentrated extract to obtain the analytical sample. A 10 μL aliquot of the analytical sample was injected into the LC/MS instrument (LC-TOF, Agilent Technologies, United Kingdom (UK)). The sample was separated with an octadecyl silica gel column (Inertsil ODS-3, GL Science, Japan; 0.5 μm × 3.0 mm × 150 mm). A formic acid–water solution (0.05% v/v) and methanol were used as mobile phases. The total flow of the mobile phases was 0.4 mL min⁻¹. The methanol fraction during each analysis was set at 10% (0 min), 90% (30 min), 90% (40 min), 10% (45 min), and 10% (60 min). The compounds from the column were analysed with the time-of-flight mass spectrometer. In our previous paper (Sato et al., 2007), recovery of malic acid ($\log_{10} C^* \approx 2$) was determined to be > 90%, suggesting that evaporation loss during pre-treatment is negligible for molecules with $\log_{10} C^* \leq 2$.

### 2.3. Dilution-induced evaporation experiment

Another 6 m³ FEP film bag (2.55 × 1.53 × 1.53 m, 50 μm) was used as an external dilution chamber (EDC). Clean air used for the dilution of SOAs was prepared using a clean air generator (Model 1160, Thermo Fisher Scientific, USA) for experiments at RH < 1%, and a purified air generator (DAR-2000, Horiba Stec Ltd., Japan) for experiments at RH ≈ 40%. The temperature of the laboratory was controlled at 298 ± 1 K. Prior to each dilution-evaporation experiment, clean air with a relative humidity identical to that in the SOA formation chamber was filled into the EDC. A necessary amount of reaction chamber air was injected into the EDC using a dilution ejector (FPS-4000, Dekati Ltd., Finland). The dilution ratio (DR) was 20–77. Dry filtered air was used as the carrier of the dilution ejector. The particle size distribution, particle density, and CO concentration in the EDC were monitored after the SOA dilution.

## 3  Results and Discussion

### 3.1. Time series

Figure 1 shows the time series of gas and particle concentrations and particle oxygen-to-carbon (O/C) ratios measured during the ozonolysis (run 8) and aging (run 9) experiments; these are shown with broken and solid lines, respectively. The results measured in run 8 are typical in simple ozonolysis experiments. The concentration of α-pinene decreased whereas the PA and SOA concentrations increased after the addition of ozone. PA is a gaseous first-generation product from α-pinene. After most of the α-pinene was consumed, PA and SOA levels became constant or decreased slowly due to loss from wall deposition in run 8. In run 9, methyl nitrite was added to the reaction mixture 58 min after the start of ozonolysis and it was then irradiated with the xenon lamps. After irradiation started, gaseous PA levels decreased and SOA levels increased due to photochemical aging. Methyl nitrite was consumed within 60 min after injection. The total OH exposure for 60 min was estimated to be (3.1–3.8) × 10¹¹ radical cm⁻³ s, based on a similar analysis described in Sato et al. (2016); this corresponds to an OH exposure of 3.6–4.4 d when the OH level is assumed to be 1.0 × 10⁶ radical cm⁻³.

The O/C ratio of SOA was ~0.40 immediately after SOA concentrations increased above detectable levels, and then decreased with time. The O/C ratio decreased to ~0.34 when most of the α-pinene was consumed and then increased slowly in run 8. Saturated organic compounds are formed from α-pinene ozonolysis; therefore, ozone-initiated reactions of the secondary products must be slow. In run 9, the O/C ratio increased to ~0.44 as a result of OH exposure, suggesting that highly oxygenated SOAs formed from photochemical aging. The results suggest that the reactions between the gaseous secondary products and OH radicals will be a major source of SOA formation during photochemical aging.

### 3.2. Chemical composition analysis

Strong signals measured during positive electrospray ionization analysis were identified as sodium-attached product ions. Although several signals were identified as protonated product ions, they were weaker than those of sodium-attached ions. Table S1 shows monomer ($C_{8-10}H_{12-18}O_{3-9}$) and dimer ($C_{16-20}H_{24-32}O_{4-12}$) products identified as sodium-attached ions. Figure 2 shows the extracted ion chromatograms (mass extraction window ±20 ppm) measured for $C_{10}H_{16}O_5Na^+$, $C_{10}H_{16}O_6Na^+$, and $C_{10}H_{16}O_7Na^+$. No sodium salt was added to the mobile phase or analytical sample. Species that do not generate stable positive ions through protonation were ionized by clustering with $Na^+$ cations that are naturally present in the solvent chemicals and glassware (Kruve et al., 2013; Zhang et al., 2017). The signal intensities of the sodium-attached ions were confirmed to have a linear relationship with the amount of injected sample. Monomers detected in a previous online study, $C_{10}H_{14-16}O_{7-11}$ (Ehn et al., 2017), are similar to those detected in our present offline analysis, suggesting that only a small portion of HOMs may decompose during pre-treatment.

$C_{10}H_{16}O_{5-7}$ compounds are HOMs with the same number of carbon and hydrogen atoms as pinonic acid. We also detected sodium-attached ions of pinonic acid ($C_{10}H_{16}O_3$) and its hydroxylated derivative ($C_{10}H_{16}O_4$), both of which are typical α-pinene SOA markers. The carbon oxidation states ($\approx$ 2 O/C - H/C; Kroll et al., 2011) of $C_{10}H_{16}O_3$, $C_{10}H_{16}O_4$, $C_{10}H_{16}O_5$, $C_{10}H_{16}O_6$, and $C_{10}H_{16}O_7$ are -1.0, -0.8, -0.6, -0.4, and -0.2, respectively. In this study, we define HOMs as molecules with carbon oxidation states of -0.6 or higher. Several peaks were observed in the chromatogram for each HOM, suggesting that several isomers are available. New peaks appeared, or peak heights increased, in chromatograms from the photochemical aging experiment (run 9; red lines) compared to the simple ozonolysis experiment (run 8; blue lines), suggesting that HOMs are produced during photochemical aging as well as ozonolysis.

Figure 3 shows the carbon number distributions measured in selected experiments; runs 1 and 6 are experiments with OH scavengers; runs 7 and 8 are experiments without OH scavengers; and run 9 is the photochemical aging experiment. Note that the abundance was directly calculated from the summation of the normalized signal intensities. We employed column-separated results to determine the distribution of products; this enabled us to remove any contributions from cluster signals originating from the mobile phases. The total area of the chromatographic peaks was calculated over retention times of 4–40 min to determine the signal abundance. Baseline signals and peaks appearing at short (< 3 min) retention times were excluded. Signal abundances were calculated for the products listed in Table S1. The abundance calculated for each product was then normalized using the total abundance of the measured products. The calculated results reflected the relative abundances of particle products, if decomposition of products during sampling and pre-treatment was negligible.

Nine- and ten-carbon products with carbon oxidation states less than -0.6 were semi-volatile products such as pinic acid and pinonic acid. These products were detected in all selected runs. Eight- to ten-carbon HOMs were also detected in all runs. The highest relative abundance of ten-carbon HOMs was observed in the photochemical aging experiment, followed in order by the experiment without OH scavengers and the experiment with OH scavengers.

These results suggest that HOMs are formed by the OH-initiated aging of α-pinene oxidation products or OH-initiated oxidation of α-pinene. HOMs were detected in the presence of OH scavengers, suggesting that HOMs are also formed from α-pinene ozonolysis. The auto-oxidation of $RO_2$ radicals (Zhang et al., 2017) or RO radicals (Müller et al., 2012) could explain present results. Both types of radicals are formed from the ozonolysis of α-pinene as well as the OH-initiated oxidation of α-pinene and its secondary products. The $RO_2$ and RO radicals produced undergo intramolecular hydrogen atom shifts, followed by successive oxidation, to form HOMs. With regard to nine- and ten-carbon HOMs, the result of run 7 was close to that of run 8. These results suggest that the relative abundances of nine- and ten-carbon HOMs in SOAs will be independent of the initial α-pinene concentration or the SOA concentration.

Dimers were detected in runs 1 and 6, suggesting that they are formed from the ozonolysis of α-pinene. Most dimers had carbon oxidation states less than -0.6; in other words, dimers were less oxygenated than monomers. If dimers are formed from the simple addition of or dehydration from monomers, their carbon oxidation states must be close to those of the precursor monomers. The results suggest that dimers will be formed from less-oxygenated monomers instead of HOMs, and that dimerization will occur prior to the formation of monomer HOMs. The results also suggest that dimers will barely be oxidized during photochemical aging, probably due to their low to extremely low volatility. Stabilised Criegee intermediates and acyl peroxy radicals are not highly functionalised because they are formed directly from α-pinene ozonolysis. Dimers may be formed from the reactions of stabilised Criegee intermediates with compounds less oxygenated than HOMs (Kristensen et al., 2014; 2016), or from the recombination of acyl peroxy radicals (Zhang et al., 2015). $C_{19}$ and $C_{20}$ dimers had the lowest abundances in run 9. These dimers may decompose during photochemical aging although the mechanism is unclear. As for the $C_{17}$ and $C_{18}$ dimers with carbon oxidation state less than -0.6, the results of run 9 was close to or slightly lower than that of run 8. These results suggest that the addition of OH radicals will barely affect or slightly reduce the relative abundances of these dimers in SOAs. The result of run 8 was higher than that of run 7, suggesting that the relative abundances of these $C_{17}$ and $C_{18}$ dimers in SOAs may increase with decreasing the initial α-pinene concentration or the SOA concentration.

Previous studies have reported the presence of terpene-derived HOMs and dimers in ambient forest particles (Ehn et al., 2014; Yasmeen et al., 2010); however, dimers are not always detected in association with pinonic acid in ambient particles (Koutchev et al., 2014). Therefore, the formation mechanisms of HOMs and dimers from α-pinene oxidation require further clarification.

### 3.3. Volatility distribution from chemical analysis

To evaluate SOA volatility distribution from LC/MS results, the saturation concentration ($C^*$) was calculated using three parameterization techniques. The first parameterization technique is a method reported by Shiraiwa et al. (2014) who parameterized molecular weight (MW) as a function of $\log_{10} C^*$ (1D function). The 1D function was fitted to saturation concentration data calculated for α-pinene oxidation products using the EVAPORATION (Estimation of VApour Pressure of ORganics, Accounting for Temperature, Intramolecular, and Non-additivity effects) model (Compernolle et al., 2011). Figure S2 plots the MW of α-pinene oxidation products as a function of $\log_{10} C^*$, and compares results obtained from the present PTR-MS measurements with the results calculated using EVAPORATION (Shiraiwa et al., 2014). Chemical structure information is necessary for predictions of saturation concentration, thus α-pinene oxidation products for which chemical structures have been previously suggested, including semi-volatile compounds, highly oxygenated molecules, and dimers, were selected for EVAPORATION calculations. We assumed that the results of parameterization were applicable to all products detected by LC/MS. EVAPORATION data were validated against the results of a different calculation method, the

SPARC online calculator (Hilal et al., 2003), and experimental results of PTR-MS. Figure 2S also shows the results calculated by Sato et al. (2016) using the SPARC online calculator. Similar results were obtained from the EVPORATION model and the SPARC online calculator, both of which are based on modified structure activity relationships. The present PTR-MS results support the predictions of the EVAPORATION model and SPARC online calculator.

The second parameterization technique is a method reported by Li et al. (2016) who parameterized $\log_{10} C^*$ as a function of the number of carbon and oxygen atoms (2D function). The 2D function was fitted to EVAPORATION data for various organic compounds including not only α-pinene oxidation products, but also other organic compounds present in the atmosphere.

The third parameterization technique is the binary fit method. In Figure S2, the linear function overestimates molecular weights in the region $\log_{10} C^* = -3$ to 0. There are significant differences between the carbon oxidation states for the products with m/z ≥300 and <300, according to our current LC/MS results. We fitted these two data sets separately for better approximation as shown in Figure S3.

The sensitivity of the electrospray ionization mass spectrometer is determined by ionization and transmission efficiencies. The ionization efficiency for sodium adduct formation has a weak correlation with the partition coefficient between vacuum and solvent, but its parameterization will not be simple due to the competition with protonated ion formation (Kruve et al., 2013). We assumed that the sodium adduct formation efficiency is constant against MW because both low- and high-MW compounds present in SOAs are mixtures of various oxygenated hydrocarbons. We assumed that the transmission efficiency of the time-of-flight mass spectrometer is proportional to $(m/z)^{1/2}$ in the region of $m/z$ 200–500 (Heinritzi et al., 2016).

Figure 4a shows the volatility distributions of gas plus particle products determined from LC/MS data using a 1D function. The signal intensity of the particle products was corrected, accounting for transmission efficiency as described above. For comparison with conventional SOA volatility data, the abundance of gas products was added to that of particle products. The former was determined based on a gas-particle partitioning model. The gas component of $\log_{10} C^* = 3$ had an abundance comparable with the particle components, whereas that of $\log_{10} C^* < 3$ had negligible abundances. The gas plus particle abundances were then normalized using total abundance. All plotted results had similar bimodal distributions consisting of semi-volatile organic compounds (with $\log_{10} C^* = 0$–3) and lower volatility organic compounds (with $\log_{10} C^* < 0$). The volatility distribution of the semi-volatile component is similar to that suggested from yield curve analysis (Lane et al., 2008), however the lower volatility component has not been predicted by traditional analysis. Organic compounds with $\log_{10} C^* < 0$ were formed even in the presence of OH scavenger, suggesting that these compounds were formed as the first-generation products from α-pinene ozonolysis. Overall volatility distribution barely changed during OH-initiated aging, although HOMs increased; this suggests that dimers determined the overall volatility distribution of α-pinene SOAs.

Figure 4b shows the results obtained using a 2D function. The volatility distributions determined by the 2D function shifted to a higher saturation concentration than those determined by the 1D function. Table S2 shows the $\log_{10} C^*$ determined for α-pinene oxidation products using the SPARC online calculator and the three parameterization methods. The $C^*$ values predicted for dimers by the 1D function agreed with the SPARC results within an error of two orders of magnitude; in contrast, the 2D function predicted $C^*$ values five to six orders of magnitude higher than the SPARC results. The accuracy of the results predicted by the 2D function was worse than those predicted by 1D function because the 2D function was fitted not only to the data of α-pinene oxidation products but also that of other organic compounds.

The volatility distributions determined using the binary fitting method (Fig. 4c) have different shapes compared to those obtained using the 1D and 2D fits, suggesting that the shape of the volatility distribution obtained by LC/MS data depends on the saturation concentration parameterization. Table S2 shows that the binary fit provides a better approximation for MBTCA than the 1D and 2D fits although the binary fit provides worse approximations for dimers than the 1D fit. Table S3 compares the average $\log_{10} C^*$ values determined for volatility distributions obtained in this study. The average $\log_{10} C^*$ values determined for the binary fit (-2.19 to -2.76) were close to or lower than those determined for 1D fit (-2.71 to -0.83) and lower than those determined for 2D fit (-0.61 to 1.44). As a sensitivity check of the transmission correction, we calculated volatility distributions obtained by a 1D fit without accounting for the transmission correction (Figure 4d). The average $\log_{10} C^*$ values determined without the transmission correction (-3.55 to -1.38) are close to or lower than those determined with the transmission correction (-2.71 to -0.83). The average $\log_{10} C^*$ values determined for all LC/MS data (-3.55 to 1.44) are lower than those determined for the yield curve (2.66), suggesting that α-pinene SOAs have a lower volatility than that expected from yield curve analysis.

### 3.4. Heat- and dilution-induced evaporation

Figure S4 shows thermograms measured during heat-induced evaporation measurements of SOAs and pinonic acid particles, where the thermogram is the plot of particle mass fraction remaining (MFR) as a function of thermodenuder temperature. The thermograms measured for α-pinene SOAs in this study were very close to previous results (e.g., Salo et al., 2011; Saha and Grieshop, 2016). The data obtained at each temperature were distributed among five runs with OH scavengers (runs 1, 2, 3, 4, and 6; mean particle diameter = 236–291 nm). Although the effects of OH scavengers, photochemical aging, and relative humidity on particle volatility were studied, the thermograms showed that all SOA results were similar to each other, within experimental uncertainties. Saha and Grieshop (2016) reported that SOA volatility increases with increasing mass concentration in the range of 5–445 μg m$^{-3}$. Although we also obtained TD-AMS data between 964 and 2,400 μg m$^{-3}$ in runs 1, 2, 3, 4, and 6 (black symbols), our results showed no clear trend with mass concentration. We surmise that the observed dispersion is probably caused by either the large variation in the present MFR data or differences in the mass concentration ranges between the current and previous studies. The variation of MFR could be caused by the variation of the residence time. Although the flow rate was not monitored during present TD-AMS measurements, the SOA concentration was monitored through the bypass as the reference. The bypass result decreased by 5–9% for each TD measurement cycle (24 min) due to the SOA loss on the chamber wall (Figure S5); however, it decreased by 17–19% for 24 min in irregular cases observed during runs 2, 3, and 4. These results suggest that there may be the variation of the residence time in these runs. The compounds comprising SOA particles had lower volatilities than pinonic acid where the $\log_{10} C^*$ of pinonic acid was determined as 2.25 by the SPARC online calculator.

Figure 4e shows the volatility distributions determined for gas plus particle products using thermogram data. Thermogram data were converted into volatility distributions using a previous empirical method developed by Faulhaber et al. (2009). In that study, the relationship between saturation concentration and the temperature at which MFR = 0.5 was measured directly for several kinds of single compound particles. We used the results of the calibration curve from Faulhaber et al. (2009) directly because the TD in our study has a residence time (~13 s at room temperature) close to that used for their TD (~15 s at room temperature). We confirmed that their calibration results are consistent with our results for pinonic acid particles. The volatility distributions determined under various conditions were again similar to each other within experimental uncertainties. The average $\log_{10} C^*$ values determined for the TD-AMS results (0.14 to 1.32) are lower than that determined for the yield curve (2.66). The

results of heat-induced evaporation also suggest that there is a substantial low- or extremely-low-volatility component in addition to the semi-volatile component. The volatility distributions in Figures 4a, 4b, 4c, and 4d

(LC/MS) have different shapes than Figure 4e (TD-AMS). The shapes of the volatility distributions obtained from LC/MS analysis may be affected by uncertainties in saturation concentration and sensitivity parameterizations as well as the existence of undetected molecules. The shape of the volatility distributions obtained from the TD-AMS may be influenced by heat-induced reactions. Figure 4f includes the volatility distribution determined from the previous TD study (Saha and Grieshop, 2016). This previous result is similar to present TD-AMS results.

Figure 5 shows the time series of the volume fraction remaining (VFR) measured for SOA particles after dilution. The number concentration and mean size of the SOA particles decreased with time. We assumed that decrease in particle numbers resulted from loss by deposition on the wall, and that the decrease in the mean diameter of the particles resulted from evaporation. To remove the influence of wall loss, VFR was determined by $r^3/r_0^3$ where $r$ is the geometrical mean size of particles and $r_0$ is the geometrical mean size immediately before dilution. The red

symbols show results obtained at a dilution ratio (DR) of 20–75 under dry conditions. Although evaporation is assumed to occur instantaneously in gas–particle equilibrium partitioning models, the VFR of SOA particles decreased very slowly and became stable more than 3 h after dilution. These results support the results reported by previous researchers (e.g. Grieshop et al., 2007; Vaden et al., 2011). Mean particle size may also have decreased due to size-dependent wall loss. The decrease in the VFR resulting from size-dependent wall loss was calculated

using directly measured size-dependent wall loss rate, and was determined to be ~2% for 3 h of measurement. The evaluated decrease in the VFR was less than the statistical error of the SMPS volume (~5%). The blue symbols are the results obtained under humid conditions. The VFR value measured 3 h after the dilution depended strongly on RH due to the changes in the viscosity or chemical composition of particles (Wilson et al., 2015; Yli-Juuti et al., 2017).

The equilibration scaling time, defined by Saleh et al. (2011; 2013), was used to characterize slow SOA evaporation. The equilibration scaling time was determined by fitting eq. 2 in Saleh et al. (2013) to the dilution data. The equilibration scaling time was determined to be 24–38 min and 33–46 min for α-pinene SOA particles generated in dry and humid experiments, respectively. The results are not compared between dry and humid experiments here because the chamber systems differ between these experiments. The current results of a 24–46

min equilibration time scale imply that the gas–particle equilibrium approximation could be applied for evaporation of α-pinene SOAs under atmospheric conditions, as suggested by previous workers (Saleh et al., 2013; Saha and Grieshop, 2016). The theoretical equilibration scaling time was also evaluated using eq. 3 in Saleh et al. (2013), where the accommodation coefficient was set to a recommended value of 0.1 for α-pinene SOAs. The theoretical equilibration scaling time was determined to be 24–41 min, which was similar to the experimental

results: 24–46 min. The mass accommodation coefficient subsumes all resistances to gas–particle partitioning other than gas phase diffusion, for example, surface accommodation, deviation from Maxwell-Boltzmann molecular velocity distribution near the particle surface, and diffusion limitations in the condensed phase (Kolb et al., 2010; Saleh et al., 2013). The mass accommodation coefficient was determined to be less than unity, suggesting that the existence of low-volatility materials in SOAs, kinetic inhibition, or some combined effect

may explain the equilibration time scale measured in this study.

The volatility distributions determined from dilution data are compared with those determined from SOA yield curves (Lane et al., 2008) as shown in Figure 4f. We determined the volatility distributions from dilution data assuming gas–particle equilibrium using the VFR data measured before and 3 h after the dilution. The normalized SOA yields obtained in dilution experiments are plotted as a function of mass concentration (Figure

S6). The volatility distributions were determined by fitting eq. 2 in Grieshop et al. (2007) to the plotted data. The

volatility distributions were only calculated in the range 1-1000 μg m$^{-3}$ because dilution data are only available in this region. The average $\log_{10} C^*$ values determined from dilution measurements (1.00–1.60) are lower than that determined from SOA yield curves (2.26).

## 4    Implication and Conclusions

In this study, we investigated the effect of OH scavenger and photochemical aging on the molecular distribution of α-pinene ozonolysis products including HOMs and dimers. By employing positive electrospray ionization combined with recent parameterization techniques for saturation concentration, we also studied product volatility distribution. To the best of our knowledge, this study is the first to analyse HOMs by column separation and to compare the product volatility distribution determined by chemical analysis with those determined by

particle evaporation measurements. The HOM detection by column separation is a valuable contribution to the current research because this technique could potentially be applied to molecular identification.

The first-generation products formed during α-pinene ozonolysis were found to include compounds less volatile than those predicted from SOA yield curves, and the equilibration time scale of dilution-induced evaporation was found to be several tens of minutes. These findings support recent results of SOA chemical

composition studies and SOA evaporation studies. In the standard VBS approach, the product volatility distributions determined by SOA yield curves are employed and no reactions are assumed to occur in the particle phase (e.g., Robinson et al., 2007; Lane et al.; 2008). Currently, only a limited number of non-standard treatments are available; e.g., Trump and Donahue (2014) took into account oligomer formation in the particle phase, and Yli-juuti et al. (2017) employed the product volatility distribution determined from dilution data.

Further improvement of the atmospheric simulation model will be necessary to explain both laboratory and ambient SOA levels.

The products of α-pinene ozonolysis were semivolatile compounds, HOMs, and dimers, and HOMs also formed during OH-initiated photochemical aging. Dimers were less oxygenated than monomers. These results are consistent with previously proposed mechanisms in which the formation of dimers is initiated by reactions of

stabilized Criegee intermediates or acylperoxy radicals, and the formation of HOMs is initiated by auto-oxidation of RO or RO$_2$ radicals.

*Author contribution.* K. S., S. R., and T. I. designed and performed chamber experiments and also conducted TD-AMS and LC/MS analyses. Y. F., A. F., and Y. K. designed and carried out dilution measurements. S. I. and H.

T. designed and carried out PTR-MS measurements. Y. M., K. T., and S. S. contributed to data interpretations from the viewpoint of atmospheric modelling. T. H., A. S., and A. T. gave technical support for TD-AMS measurements and also contributed to data interpretations.

*Acknowledgements.* The authors were supported by the Environmental Research and Technology Development

Fund from the Ministry of the Environment, Japan (5-1408, FY2014-2016). K. S. and Y. M. were supported by JSPS KAKENHI Grant Numbers JP25340021, JP16H06305, and JP17H01866. S. I. and H. T. were supported by the Steel Foundation for Environmental Protection Technology (14−15 Taiki-221) and the Sumitomo Foundation (No.123449). The authors would like to thank Akio Togashi of Randstad Co., Yutaka Sugaya of National Institute for Environmental Studies, and Tsuyoshi Fujii of Horiba Techno Service Co. for their technical supports to

measurements of dilution-induced evaporation. The authors thank Andrew Grieshop of North Carolina State

University and the anonymous reviewer for useful suggestions. K. S. thanks Yoshikatsu Takazawa for technical supports to LC/MS analysis.

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

Table 1. Experimental conditions, particle mass concentration, and mean particle size.

| run | reaction [a] | RH | $[HC]_0$ | $[O_3]_0$ | $[SOA]$ [c] | size | measurements [d] |
|---|---|---|---|---|---|---|---|
| | | % | ppmv | ppmv | $\mu g\ m^{-3}$ | nm | |
| 1 | α-pinene/$O_3$/$Et_2O$ | <1 | 0.84 | 0.95 | 2,400 | 288 | TD-AMS, LC/MS, EDC (DR = 75) |
| 2 | α-pinene/$O_3$/$Et_2O$ | <1 | 0.54 | 1.02 | 1,539 | 262 | TD-AMS, EDC (DR = 20) |
| 3 | α-pinene/$O_3$/$Et_2O$ | <1 | 0.51 | 1.08 | 1,382 | 236 | TD-AMS, EDC (DR = 43) |
| 4 | α-pinene/$O_3$/$Et_2O$ | <1 | 0.53 | 1.13 | 1,490 | 291 | TD-AMS |
| 5 | α-pinene/$O_3$ | <1 | 0.11 | 0.58 | 216 | 198 | TD-AMS, PTR-MS |
| 6 | α-pinene/$O_3$/CO | <1 | 0.31 | 0.65 | 964 | 257 | TD-AMS, LC/MS, PTR-MS |
| 7 | α-pinene/$O_3$ | <1 | 0.32 | 0.59 | 859 | 274 | TD-AMS, LC/MS, PTR-MS |
| 8 | α-pinene/$O_3$ | <1 | 0.15 | 0.62 | 303 | 226 | TD-AMS, LC/MS, PTR-MS |
| 9 | α-pinene/$O_3$ (+ aging [b]) | <1 | 0.16 | 0.53 | 461 | 233 | TD-AMS, LC/MS, PTR-MS |
| 10 | α-pinene/$O_3$/$Et_2O$ | ~40 | 0.46 | 1.09 | 1,593 | 259 | EDC (DR = 33) |
| 11 | α-pinene/$O_3$/$Et_2O$ | ~40 | 0.46 | 1.09 | 1,947 | 298 | TD-AMS, EDC (DR = 20) |
| 12 | α-pinene/$O_3$/$Et_2O$ | ~40 | 0.46 | 1.09 | 1,671 | 260 | EDC (DR = 77) |

[a] Diethyl ether ($Et_2O$) and carbon monoxide (CO) were used as OH scavengers. [b] SOA formed from ozonolysis was exposed to OH radicals. [c] Calculated by assuming particle density to 1.34 g $cm^{-3}$ (present study). [d] EDC is external dilution chamber; DR is dilution ratio.

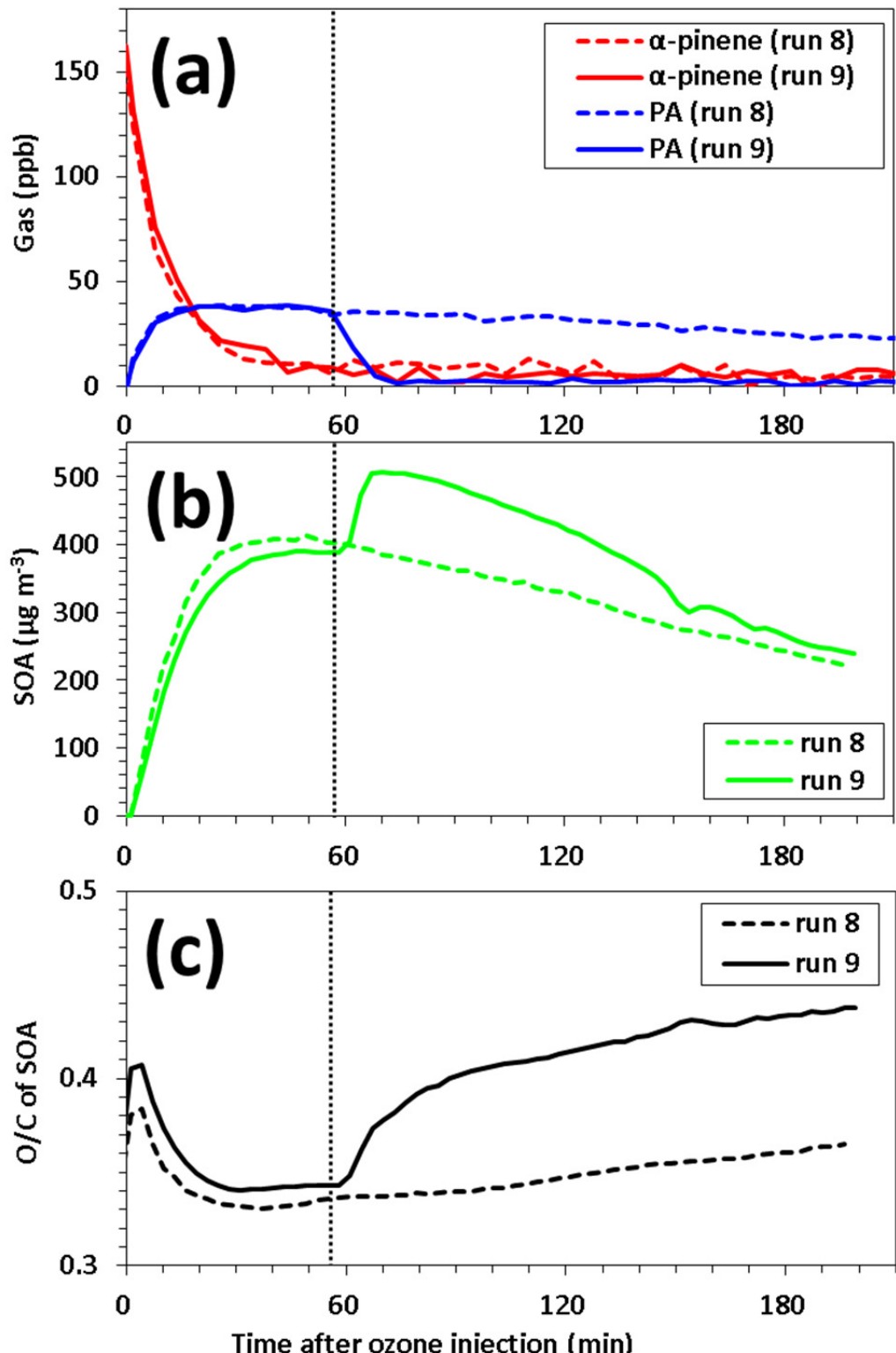

**Figure 1: Time series of (a) FT-IR gas concentrations and (b) AMS SOA concentration and (c) AMS SOA O/C ratio during simple ozonolysis experiment (run 8) and aging experiment, in which SOA formed from ozonolysis was exposed to OH radicals (run 9).**


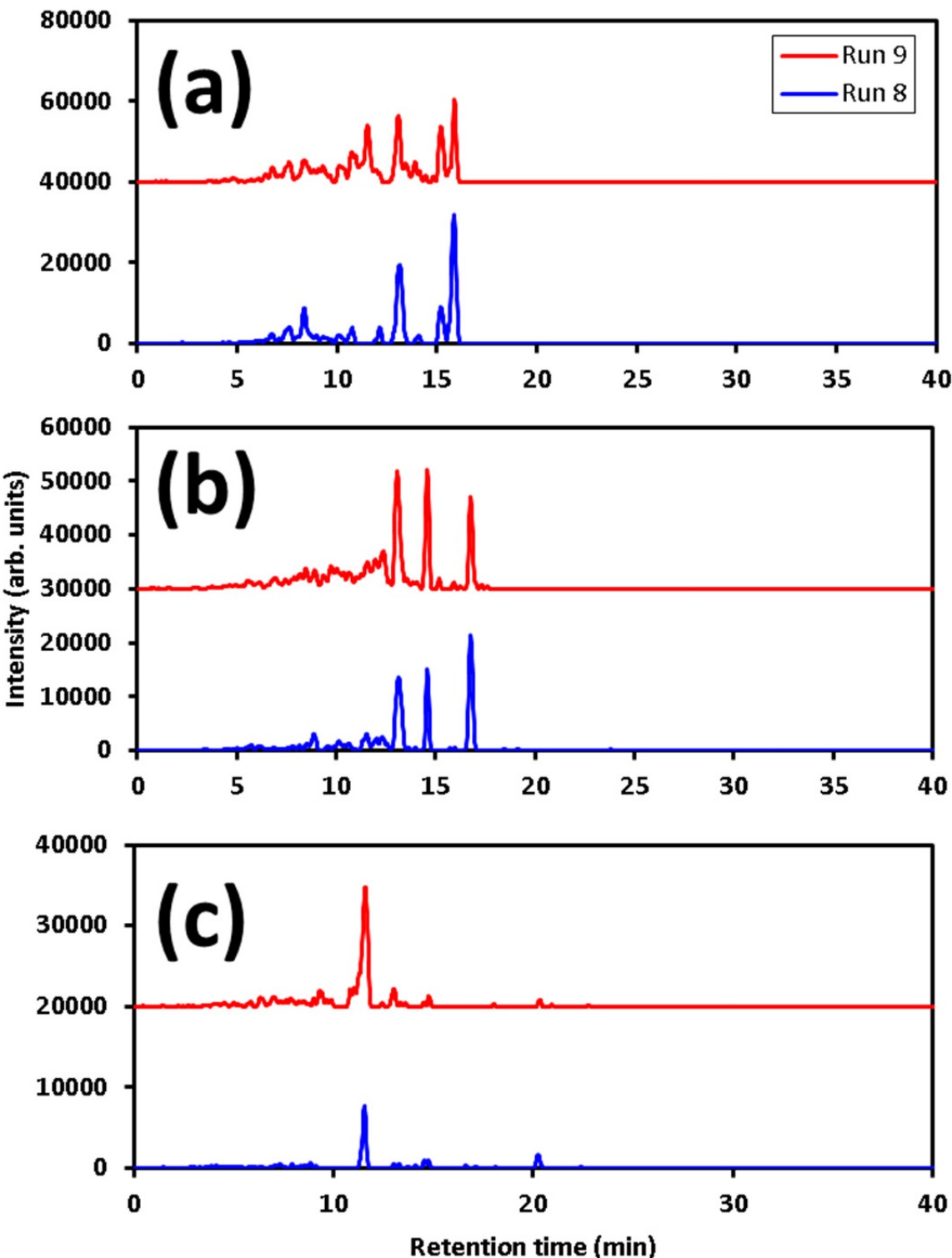

**Figure 2:** Extracted ion chromatograms of (a) $C_{10}H_{16}O_5Na^+$, (b) $C_{10}H_{16}O_6Na^+$, and (c) $C_{10}H_{16}O_7Na^+$ formed by the addition of sodium ion during the electrospray ionization of highly oxygenated molecules present in fresh (run 8) and aged SOA particles (run 9).


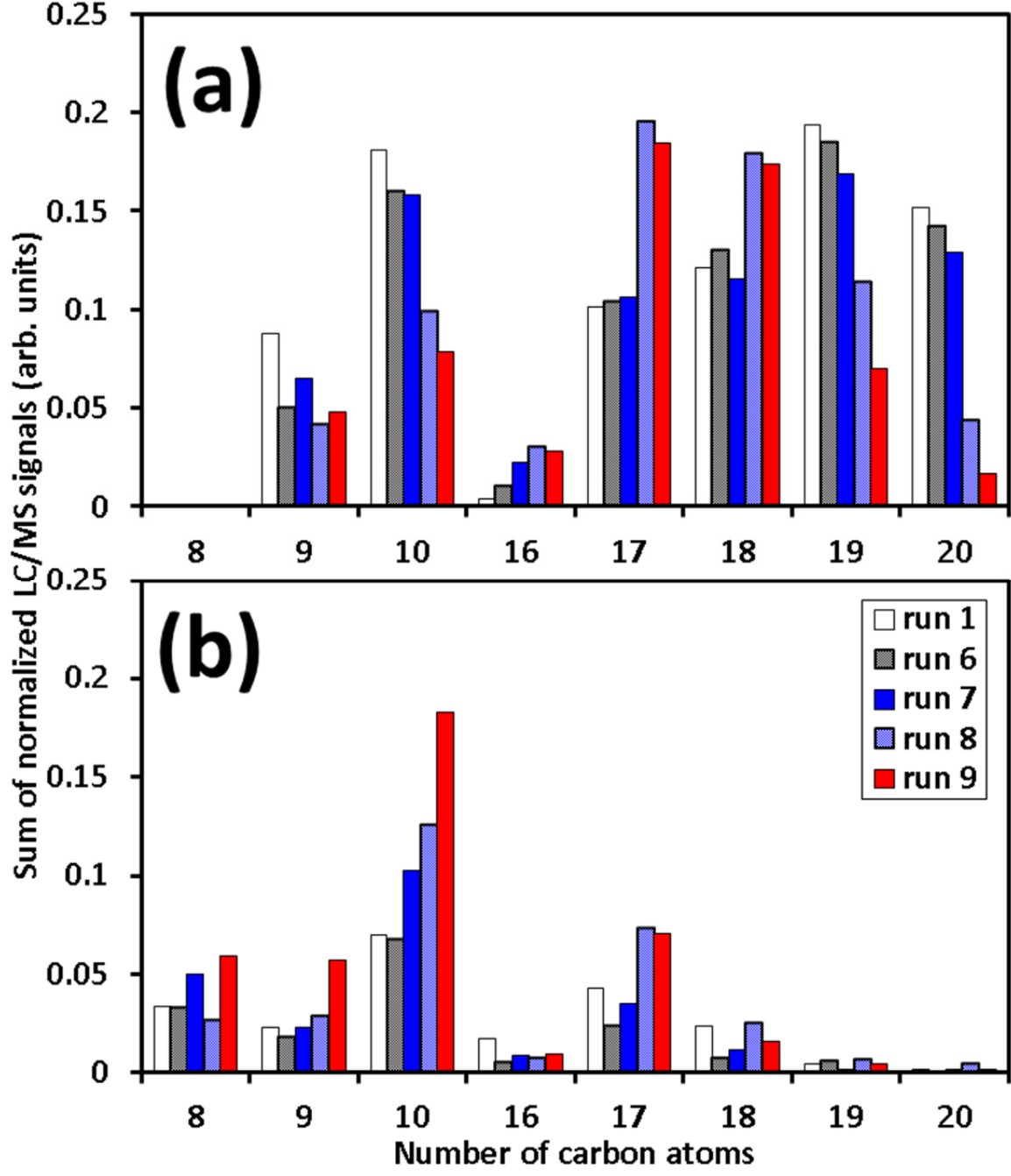

**Figure 3: Carbon number distributions determined for particle products at (a) carbon oxidation state < -0.6 (less oxygenated than HOMs) and (b) carbon oxidation state ≥ -0.6 (HOMs); abundance was directly calculated from the summation of normalized signal intensities.**

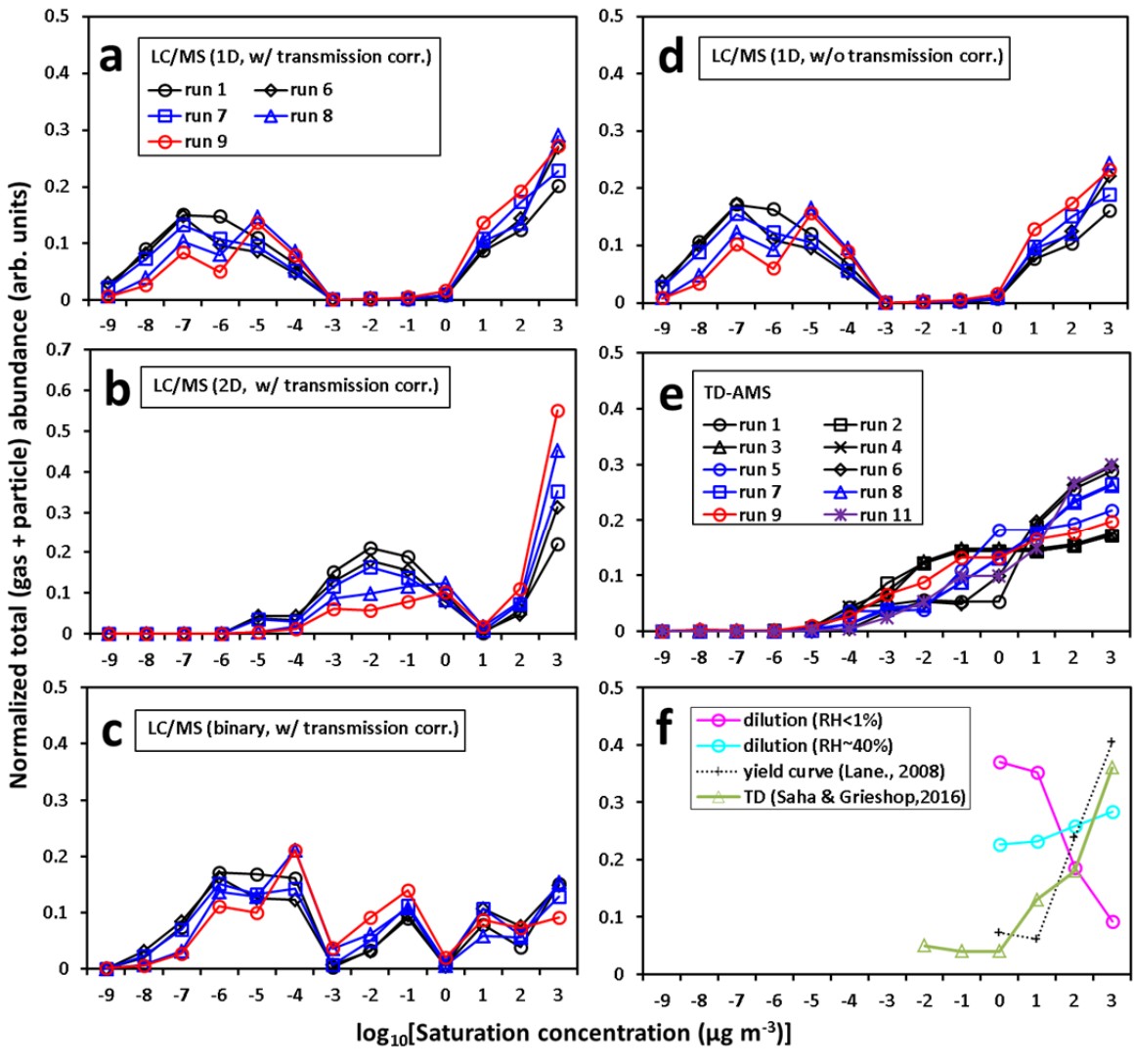


**Figure 4: Volatility distributions determined for gas + particle products from (a) LC/MS data (1D fit), (b) LC/MS data (2D fit), (c) LC/MS data (binary fit), (d) LC/MS data (1D fit, no transmission correction), (e) TD-AMS data, and (f) dilution data; The results of dilution data are compared with the volatility distribution determined from SOA yield curves (Lane et al., 2008) and TD measurements (Saha and Grieshop, 2016).**

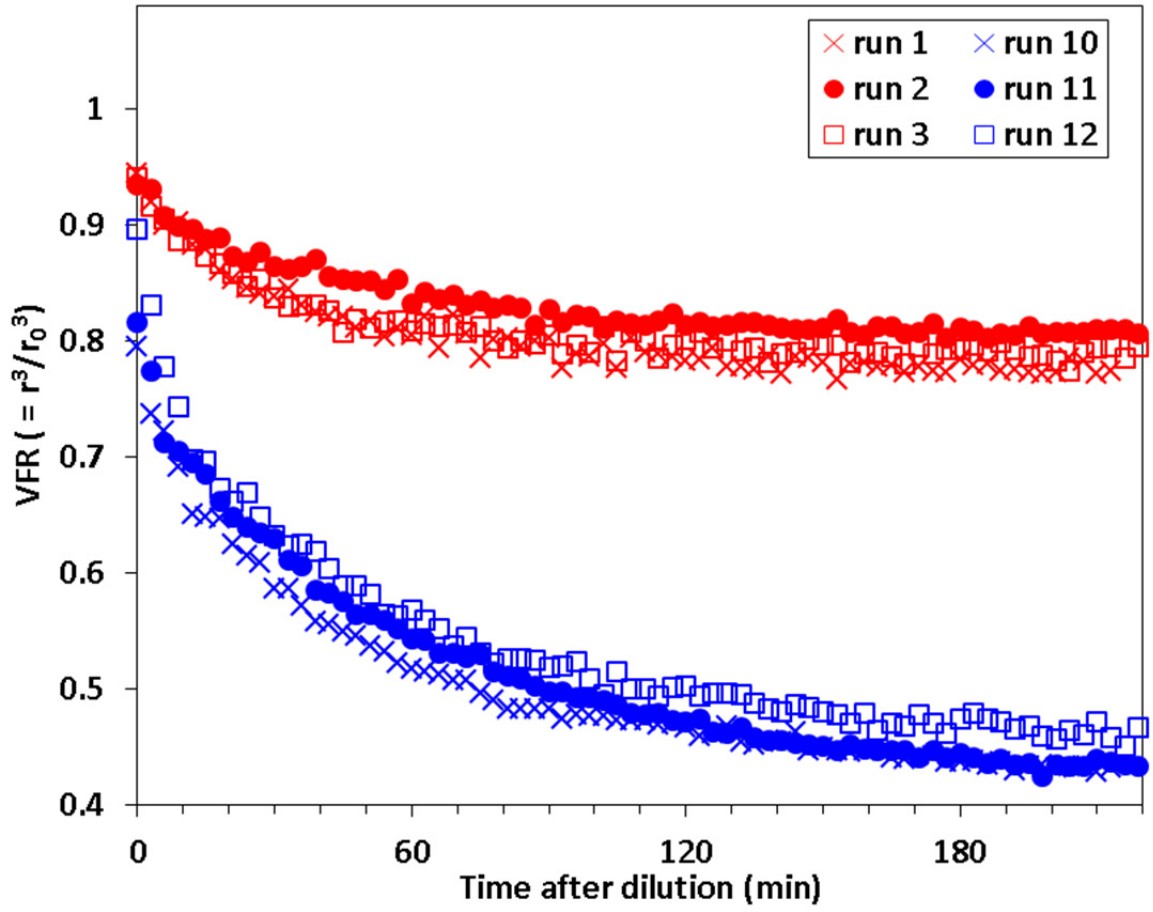


**Figure 5: Time series of volume fraction remaining (VFR) measured for SOA particles after dilution under dry (RH <
1%; runs 1, 2, and 3) and humid conditions (RH ≈ 40%; runs 10, 11, and 12).**