# Peer review of "Studying volatility from composition, dilution, and heating measurements of secondary organic aerosols formed during $\alpha$ -pinene ozonolysis"

_Atmospheric Chemistry and Physics, 2017_

## Referee Comment (RC1) · Anonymous Referee #1 · 24 Oct 2017

The authors investigated volatility of secondary organic aerosol (SOA) from a-pinene oxidation using measurements of chemical composition of particles, evaporation upon heating and evaporation upon dilution. The topic is important in the field of atmospheric aerosols and well suited in Atmospheric Chemistry and Physics. The experimental work coupling different methods for investigating the volatility of SOA seems valid. I find that there are few aspects in the analysis and discussion of the results which should be improved/clarified before the manuscript can be published. Please find below my specific comments.

Specific comments:

1. How large fraction of compounds are actually detected with PTR-MS? Is sticking of molecules on the filter affecting uncertainty of the resulting Csat?

2. Particles were formed in different chambers in dry and 40% RH case: one with Teflon-coated walls and the other made of fluorinated ethylene polyethylene. At least Teflon walls are found previously to take up organic vapors. This can affect the composition (i.e. volatility distribution) of the particles and reduce the comparability of the dry and 40% RH experiments. Also, the SOA formation conditions have not been same in both types of experiments. The authors should discuss the effect of such possible sources of uncertainty in context of comparing the dry and 40% RH experiments. Also, please make it clear what was the RH in the evaporation section of the set-up when the particles formed at 40% RH were studied. I assume it was 40% (P4, L126).

3. LC/MS measurements of composition: A. Can the SOA evaporate or react during the treatment of the filter sample? How much uncertainty does this cause to the measured composition? Especially, one would expect some of the semi-volatiles to evaporate when the filter extract is concentrated in nitrogen stream. If such effects are possible, the effect on the inferred SOA composition and VBS should be discussed. B. How large fraction of SOA compounds are expected to be detected with the method and do the compounds that are not detected cause uncertainty to the results? The ionization and transmission efficiencies are discussed on page 6 related to Csat distribution. Were the same efficiencies assumed when analyzing carbon number distributions? C. Based on Figure 3 there are rather large differences between runs 1 and 6 and between 7 and 8 although the oxidation conditions have been similar. This should be addressed in the discussion of the results.

4. Authors present the Csat distributions as sum of particle and gas phase (Figure 4) where particle phase is based on measured composition and gas phase is calculated assuming gas-particle equilibrium. Based on Figure 3, O/C is continuing to change at the end of the experiment suggesting that the system is not in equilibrium. Based on text on page 6 it's not clear if this was considered in the analysis when estimating abundance of gas phase compounds.

5. The volatility distributions in Fig. 4 a-b have different shape compared to Fig. 4 c.

Do authors have any ideas on what causes this?

6. Heat induced evaporation (Figure S3 and P7 L248-260): The variation of MFR at T=50C seem rather large for cases with OH scavenger (black markers). Did the authors look into what could cause this variation? Authors state that "Although the effects of OH scavengers, photochemical aging, and relative humidity on particle volatility were studied, the thermograms showed that all SOA results were similar to each other, within experimental uncertainties." Can the variation between the black markers be something else than experimental uncertainty? In that case there could be some effect of OH scavenger, aging and RH hidden in the data points.

7. "The present results indicate that gas/particle partitioning was virtually irreversible even though the VFR continued to decrease after 3 h." (P7, 282) This is unclear. What do the authors mean by irreversibility here?

8. "To the best of our knowledge, this study is the first to arrive at the results described above." (P8, L291) Please be more specific here. Not all the results described in the manuscript are as new as this suggests. Highly oxidized and/or long carbon chain compounds and slow evaporation are detected in earlier studies also, e.g. in the references of the current manuscript.

9. Please consider revising the title "Lower than expected volatility of secondary organic aerosols formed during $\alpha$-pinene ozonolysis". The evaporation of particles was indeed slower than expected based on the yield experiments. However, considering Vaden et al. (2011, ACP) and other studies since then (which authors reference in the manuscript), I don't think "lower than expected" is anymore valid. With this I do not by any means suggest that the current study would not be important. It is definitely a topic which is worth of extensive research. However, the current title suggests that the authors find even lower volatility than Vaden et al. and other studies have found, which doesn't seem to be the case.

10. Figure S2 and Table S2. These include only a small number of compounds and

different compounds for different methods. It is not clear why these specific compounds were chosen for these comparisons.

Technical comments:

P1, L22: "Atmospheric fine aerosols are believed to negatively affect climate (IPCC, 2013)..." Is this a typo? What do authors mean by 'negatively affect climate'?

P1, L39: "Wilson et al., 2014" should probably be 2015.

P2, L41-43: "...however, later research showed that the evaporation process is strongly influenced by the particle phase state, and suggested that dilution results cannot be simply interpreted by gas/particle partitioning." Please add reference to this statement.

P5, L179: "The highest relative abundance was observed in the photochemical aging experiment,..." Please specify which compounds you refer to.

P5, L191: "...that such dimerization will occur prior to the formation of the latter." Please clarify this sentence.

P7, L271: "Although evaporation is assumed to occur instantaneously in VBS models,..." Did you mean equilibrium partitioning models? VBS is often used also for representing dynamics of condensation/evaporation.

P8, L297: "...and photochemical aging in gas phase would be slower than that expected in the model." Please clarify this sentence. It is not clear how this conclusion was reached.

Figure S2. At the start of figure caption, please change "MW" to "Molecular weight (MW)" for clarity.

---

## Referee Comment (RC2) · AP Grieshop (Referee) · 11 Nov 2017

Note: I am a referee for the paper but am posting my review with my name as much of my response is informed by a highly related (and complementary) paper that a former student and I published on a similar topic (Saha and Grieshop 2016). It appears that the authors of this paper may not be aware of our paper, and I wished to be transparent so thought it most appropriate to post this review under my name.

This paper presents results from a series of chamber experiments in which a-pinene was oxidized with ozone and OH and the resulting aerosol measured in-situ and via measurements of particulate matter. A-pinene ozonolysis is a well-studied system, but a number of open questions remain, with the product distribution (typically quanti-

fied in terms of volatility) and phase-partitioning kinetics being among two of the most pressing ones. This paper adds to the literature discussing this system by including cutting-edge chemical analyses of particles formed from both ozonolysis and particles thus formed and then perturbed via OH oxidation and additional dilution. A number of different experimental arrangements – e.g. use of thermodenuder (TD), analysis of PTR-MS measurements of gas-phase and vapors driven off a filter, ESI-MS analysis of filter samples and multi-chamber dilution studies – provide valuable data sets by which the partitioning and chemical composition of this SOA can be probed. This is a system and a topic of great interest to ACP readers, and as such I believe this is highly appropriate venue for this analysis. This is an interesting and creative set of experiments resulting in data collected with cutting edge techniques. However, while the data appear to be carefully collected, I find some of the interpretation of these data could be improved and that some of the resulting conclusions are insufficiently supported by the analysis. Therefore, I suggest that this paper be substantially revised before publication is considered.

A number of specific points are included below, but the main concerns I have are:

1) The title and tone of discussion don't give proper weight to previous work that has shown that the partitioning of a-pinene SOA is not well-represented by 'traditional' yield experiments. As the other reviewer suggested, this is not a 'new' insight, and so more effort should be made to put the present results into the proper context, including that provided by our paper and others with more detailed chemical analyses that well preceded ours.

2) Inter-experiment variation is an important theme that should be explored. There is relatively little comparison between sets of experiments conducted at similar conditions to establish repeatability of efforts (which seem to be good for TD results, based on Runs 2, 3 and 4 in Fig. 4c) and also to more systematically explore the influence of different experimental conditions (e.g. OH scavenger, RH) on the measurements. One complication, especially to interpreting TD results is that OA concentration varies quite

substantially (Table 1). Our paper showed that smog chamber OA concentration has a systematic effect on its observed evaporation. These data are collected at a range of concentrations, though a number of other parameters were simultaneously changed and so it's difficult to say whether the effect of individual changes can be extracted. In addition to inter-experimental variability, I'm also curious about possible explanations for the quite substantial variation in volatility distributions from different methods (Fig. 4 panels) – are there any cases in which experiments show better or worse agreement? What might explain these differences?

3) I have some major concerns about the collection and analysis of the room-temperature evaporation data. In considering these results, I'd ask the authors to take a careful look at our paper (Saha and Grieshop 2016) and also work by Saleh et al. (Saleh et al. 2011, 2013), especially their chamber experiments considering time scales of equilibration. An important point is that equilibration rates cannot be readily separated from volatility in this type of experiment, and so the authors should not compare the assumed equilibrium of particles calculated based on yield experiment data (horizontal lines in Fig. 5) with dynamic equilibration data unless the consistency is somehow proven (e.g. the actual equilibrium state of the system is known). This disconnect especially stands out here because the authors have presented volatility distributions showing much lower volatility, so it is strange to compare 'equilibrium' dictated by another experimental approach (yield experiments) when other parts of your paper are essentially arguing that these are wrong. A better way to compare equilibrium properties is by comparing volatility distributions. Normalized time scales (see Saleh et al. papers referenced above) can be used to compare equilibration time scales for such experiments. As it is, this figure suggests that equilibration time scales for the low and high RH experiments are very different, but this difference may be due to differences in volatility or other experimental conditions. If this is the case, the volatility distributions in Fig. 4 may be able explain this difference. As both our paper and (Saleh et al. 2013) point out, if you assume the wrong volatility you can easily conflate 'slow' evaporation due to a kinetic limitation (as you suggest), when it's really just that

(as your chemical analyses suggest) the aerosol is much less volatile than yield data suggest. This is an assumption that we show may be at play in the 'kinetic limitation' ascribed to slow evaporation by Vaden et al. (2011). In addition, since different chambers (Teflon-coated steel vs. FEP membrane) were used for the different experiments, it's impossible to say what influence different wall characteristics may have of on the observed evaporation. As it is, there is far too little evidence to support the statements on Line 279-284 about 'irreversible' partitioning and the RH-dependence of evaporation and viscosity. There is good evidence elsewhere that RH dependencies may be important, but I don't find what is presented here convincing. For example, the data don't support the statement (L282-283) that that 'gas/particle partitioning was virtually irreversible'. This level of dilution is insufficient to probe the irreversibility of partitioning for compounds with $C^*$ less than $10^1$ or $10^2$ ug/m3, which comprise the majority of the material in your distribution (Fig. 4) but not the Lane et al. (2008) distribution used to calculate equilibrium partitioning.

Specific points

L42-43 – Discussion of dilution results is limited, and the body of literature lumped as 'later research' should be further discussed. This is a logical place to discuss Saleh et al. (2013).

L90 – It would be helpful to compare density numbers determined here with those from other studies. In general, more comparison of results with the wealth of other studies of this model aerosol system would help to place this study in its proper context relative to the literature.

L97-98 - Discussion of the use of pinonic acid as a reference for the thermodenuder (TD) measurements should be discussed, along with more generally the approach used for interpreting TD measurements to yield Fig 4c. Is the T_50 method of Faulhaber et al. (2009) used with only a single 'calibration' compound (pinonic acid) to develop the calibration? Or were more calibration compounds used? Was the calibration from Faulhaber used directly? If so, do the results for your single calibration compound match well with that from their calibration? It is not clear if your TD has exactly the same residence time as the one used there, which would be an important pre-requisite for applying this calibration.

Line 102-104 – I would be curious to see how wall losses of vapors might affect the determination of the distribution of gas-phase constituents in the chamber at the end of the experiment and how this, and also sampling/storage conditions might affect what was sampled on the chamber walls. Also, during the desorption of the filters, was the PTR-MS signal observed to return to background? In both your and our work, we find materials that remain in the particle phase (not on filters) in a TD to up to 120 C (393 K) with substantial residence times, so I'd be curious to see how PTR-MS-identified products evolve during desorption and if everything was actually desorbed from the filters. In general, no PTR-MS data are shown apart from the points shown in Fig. S2 in the supplement. Is there any reason these data are not further discussed?

Line 109 and 152-153 – A brief description of the analytical method would be helpful. For example, some mention of the source/significance of sodium adducts would be helpful, especially for those (like me) unfamiliar with this quirk of ESI analysis.

Line 241 – Error of +/- 2 – it is not clear what units this error is reported in? Possibly a factor of 2?

Line 212-215 – While a line can be drawn through these data points, and shows a reasonably high R^2 value, there seems to be a discontinuity in this relation at a logC* of around -3 or -4, with the clusters of data at either side of this showing a much shallower relationship between MW and logC*. For example, if you exclude the lower volatility data, you would have a much different dependence. How does this affect your results? Is there any reason for this discontinuity, perhaps due to the oxidation state or other properties of the compounds in these two clusters?

L221-223 – This may be a very normal assumption to make for a TOF-MS, but I would

like to see it justified, either with a reference or calibration data. How sensitive are results (e.g. volatility distributions) to any uncertainty in this?

L242-244 – I don't quite understand this statement. How were 2D function results 'adjusted'? I'm assuming this is referring to the range of compounds fit by the 2D function? This statement and its implications should be clarified.

Line 250-255 – The level of agreement should be better described (perhaps by comparing others' data to yours). Also, I note that there is quite a bit of scatter in observed evaporation, especially at 50 deg. C, where there is nearly a factor of 2 range in VFR ($\sim$0.4 to 0.7) for the alpha-pinene SOA systems. It is stated that data were similar 'to within experimental uncertainties', but these uncertainties are not stated or discussed. As noted above, comparing results collected for experiments at the same conditions (Coa, scavenger, RH, etc.) can constrain inter-experiment variability and then be used to discuss whether any observed differences may be ascribed to these conditions or uncontrolled variability in the experiments.

Line 291 – Our study and various others that our paper discussed and cited , many of which were also cited here (e.g. (Ehn et al. 2014; Zhang et al. 2015)) have pointed towards the prevalence of very low volatility compounds in pinene SOA in contrast to existing yield-experiment-based parameterizations. It may be true that you are the first who arrived at the results via application for ESI-MS, but that is not how I read this sentence.

Line 293-294 – Since a VBS model by itself doesn't assume an evaporation rate, it's hard to make this comparison. Also, as noted, volatility and evaporation kinetics are conflated in the comparison in this paper, so insufficient evidence is presented to make this claim. In chemical transport models, the assumption is that frequently made that equilibrium partitioning adequately describes what happens within time steps of on the order of a $\sim$hour. Our paper argues that this is probably a fine assumption for alpha-pinene SOA produced in our lab under atmospheric conditions, though the assumption

cannot be made in interpreting thermodenuder measurements.

References

Ehn, M., Thornton, J. A., Kleist, E., Sipilä, M., Junninen, H., Pullinen, I., Springer, M., Rubach, F., Tillmann, R., Lee, B., Lopez-Hilfiker, F., Andres, S., Acir, I.-H., Rissanen, M., Jokinen, T., Schobesberger, S., Kangasluoma, J., Kontkanen, J., Nieminen, T., Kurtén, T., Nielsen, L. B., Jørgensen, S., Kjaergaard, H. G., Canagaratna, M., Maso, M. D., Berndt, T., Petäjä, T., Wahner, A., Kerminen, V.-M., Kulmala, M., Worsnop, D. R., Wildt, J., and Mentel, T. F. (2014). "A large source of low-volatility secondary organic aerosol." Nature, 506(7489), 476–479.

Faulhaber, A. E., Thomas, B. M., Jimenez, J. L., Jayne, J. T., Worsnop, D. R., and Ziemann, P. J. (2009). "Characterization of a thermodenuder-particle beam mass spectrometer system for the study of organic aerosol volatility and composition." Atmospheric Measurement Techniques, 2(1), 15–31.

Lane, T. E., Donahue, N. M., and Pandis, S. N. (2008). "Simulating secondary organic aerosol formation using the volatility basis-set approach in a chemical transport model." Atmospheric Environment, 42(32), 7439–7451.

Saha, P. K., and Grieshop, A. P. (2016). "Exploring Divergent Volatility Properties from Yield and Thermodenuder Measurements of Secondary Organic Aerosol from $\alpha$-Pinene Ozonolysis." Environmental Science & Technology, 50(11), 5740–5749.

Saleh, R., Donahue, N. M., and Robinson, A. L. (2013). "Time Scales for Gas-Particle Partitioning Equilibration of Secondary Organic Aerosol Formed from Alpha-Pinene Ozonolysis." Environmental Science & Technology, 47(11), 5588–5594.

Saleh, R., Shihadeh, A., and Khlystov, A. (2011). "On transport phenomena and equilibration time scales in thermodenuders." Atmospheric Measurement Techniques, 4, 571–581.

Vaden, T. D., Imre, D., Beránek, J., Shrivastava, M., and Zelenyuk, A. (2011). "Evapo-

ration kinetics and phase of laboratory and ambient secondary organic aerosol." Proceedings of the National Academy of Sciences.

Zhang, X., McVay, R. C., Huang, D. D., Dalleska, N. F., Aumont, B., Flagan, R. C., and Seinfeld, J. H. (2015). "Formation and evolution of molecular products in $\alpha$-pinene secondary organic aerosol." Proceedings of the National Academy of Sciences of the United States of America, 112(46), 14168–14173.

---

## Author Comment (AC1) · 22 Jan 2018

**Reply to Referee #1**

Kei Sato, Yuji Fujitani, Satoshi Inomata, Yu Morino, Kiyoshi Tanabe, Sathiyamurthi Ramasamy, Toshihide Hikida, Akio Shimono, Akinori Takami1, Akihiro Fushimi, Yoshinori Kondo, Takashi Imamura, Hiroshi Tanimoto, Seiji Sugata

**General comments**

*The authors investigated volatility of secondary organic aerosol (SOA) from a-pinene oxidation using measurements of chemical composition of particles, evaporation upon heating and evaporation upon dilution. The topic is important in the field of atmospheric aerosols and well suited in Atmospheric Chemistry and Physics. The experimental work coupling different methods for investigating the volatility of SOA seems valid. I find that there are few aspects in the analysis and discussion of the results which should be improved/clarified before the manuscript can be published. Please find below my specific comments.*

**Reply**

Thank you for your valuable comments; we appreciate the time and effort you have put into reviewing our paper. I have revised the manuscript based upon your input, as described below.

**Specific comments**

**Comment 1**

*How large fraction of compounds are actually detected with PTR-MS? Is sticking of molecules on the filter affecting uncertainty of the resulting Csat?*

**Reply**

We compared the mass concentration of gaseous products detected by the PTR-MS ($\Delta[\text{Gas}]_{\text{PTR-MS}}$) with the expected value, which was calculated from a difference between the amounts of consumed reactants and the SOAs formed ($\Delta[\text{M}]_c - [\text{SOA}]$). The results of runs 5, 6, 7, and 8 are summarized in Table R1. In run 6, for example, 0.31 ppmv of $\alpha$-pinene and 0.27 ppmv of $O_3$ were consumed after a reaction time of 1 h, which corresponds to 2.3 mg m$^{-3}$ in total. The amount of gaseous products detected by the PTR-MS was estimated to be 1.0 mg m$^{-3}$, using a calculated detection sensitivity (9.1 ncps/ppbv in this study) with a typical ion-molecular reaction rate constant ($2 \times 10^{-9}$ cm$^3$ molecule$^{-1}$ s$^{-1}$) for all products. Since the amount of SOAs was 1.0 mg m$^{-3}$, the amount of gaseous products detected by the PTR-MS accounts for 77 % of ($\Delta[\text{M}]_c - [\text{SOA}]$). Except for run 7, gaseous products detected by the PTR-MS were 60−77% of the expected value ($\Delta[\text{M}]_c - [\text{SOA}]$).

The amount of products in the aerosol phase was estimated by summing ion signals of evaporated compounds during the heating of the sample filter. The results for runs 6 and 7 are listed in Table R1. About 30 % of SOAs measured by SMPS was detected by the PTR-MS.

We checked the time variations of the PTR-MS signals of m/z = 185 (pinonic acid) and 187 (pinic acid) after the aerosol sample filter was heated at 368 K (please see the specific comment 4 made by the other reviewer and Figure R2). The signals of m/z = 185 and 187 decreased to very low levels within 3 h, suggesting that the heating duration will be sufficient. Even though the PTR-MS signals decreased to zero after the heating, these compounds might be still adsorbed on the filter media. It might be difficult to determine the amount of compound remaining on filter. If 10% of the collected compound is not evaporated from the filter, we overestimate Csat and the result of $\log_{10}$ Csat shifts positively by 0.05 from that obtained when all compound is evaporated. If 20% of the collected compound is not evaporated, the result of $\log_{10}$ Csat shifts positively by 0.10. As far as we discuss the order of magnitude of Csat, the influence from sticking molecules will be small.

**Comment 2**
*Particles were formed in different chambers in dry and 40% RH case: one with Teflon-coated walls and the other made of fluorinated ethylene polyethylene. At least Teflon walls are found previously to take up organic vapors. This can affect the composition (i.e. volatility distribution) of the particles and reduce the comparability of the dry and 40% RH experiments. Also, the SOA formation conditions have not been same in both types of experiments. The authors should discuss the effect of such possible sources of uncertainty in context of comparing the dry and 40% RH experiments. Also, please make it clear what was the RH in the evaporation section of the set-up when the particles formed at 40% RH were studied. I assume it was 40% (P4, L126).*

**Reply**
We agree with your comments on potential differences in experimental conditions between experiments under dry and humid conditions. We discuss possible uncertainty in the context of comparing the dry and humid experiments in lines 336-337 in the revised manuscript as follows:

The results are not compared between dry and humid experiments here because the chamber systems differ between these experiments.

Your assumption on the RH of evaporation is correct. The RH of EDC was set to ~40% when particles formed at ~40%, whereas it was set to <1% when particles formed at <1%. We explain this in lines

136-138 of the revised manuscript as follows:

Prior to each dilution-evaporation experiment, clean air with a relative humidity identical to that in the SOA formation chamber was filled into the EDC.

We also explain this in the caption of Figure S1 of the supplementary material.

**Comment 3**

*LC/MS measurements of composition: A. Can the SOA evaporate or react during the treatment of the filter sample? How much uncertainty does this cause to the measured composition? Especially, one would expect some of the semi-volatiles to evaporate when the filter extract is concentrated in nitrogen stream. If such effects are possible, the effect on the inferred SOA composition and VBS should be discussed. B. How large fraction of SOA compounds are expected to be detected with the method and do the compounds that are not detected cause uncertainty to the results? The ionization and transmission efficiencies are discussed on page 6 related to Csat distribution. Were the same efficiencies assumed when analyzing carbon number distributions? C. Based on Figure 3 there are rather large differences between runs 1 and 6 and between 7 and 8 although the oxidation conditions have been similar. This should be addressed in the discussion of the results.*

**Reply**

A. We have added following text at lines 171–173 to discuss the effect of reactions during pre-treatment:

Monomers detected in a previous online study, $C_{10}H_{14-16}O_{7-11}$ (Ehn et al., 2017), are similar to those detected in our present offline analysis, suggesting that only a small portion of HOMs may decompose during pre-treatment.

Furthermore, we added the following sentence in lines 129–131 to discuss the effect of evaporation during pre-treatment:

In our previous paper (Sato et al., 2007), recovery of malic acid ($\log_{10} C^* \approx 2$) was determined to be > 90%, suggesting that evaporation loss during pre-treatment is negligible for molecules with $\log_{10} C^* \leq 2$.

B. If we apply pinonic acid-equivalent sensitivity for all product signals, the total mass of detected products will be estimated to be 153–273% of the total particle mass; suggesting that the sensitivity of pinonic acid is lower than an effective sensitivity for the total products. Because we do not know

the effective sensitivity for the total products, we cannot evaluate how large a fraction of SOA compounds is expected to be detected with the method. We agree with the comment. Undetected compounds would affect the volatility distributions determined from LC/MS data. We have added the following sentences in lines 314–317 of the revised manuscript:

The volatility distributions in Figures 4a, 4b, and 4c (LC/MS) have different shapes than Figure 4e (TD-AMS). The shapes of the volatility distributions obtained from LC/MS analysis may be affected by uncertainties in saturation concentration and sensitivity parameterizations as well as the existence of undetected molecules.

When we calculated carbon number distributions, we plotted the sum of raw signal intensities without using any efficiency (this is written in the caption of Fig.3). We revise the vertical axis title of this figure to "Sum of LC/MS signals (arb. units)". Furthermore, we add the following sentence in line 185 in the manuscript.

Note that the abundance was directly calculated from the summation of the signal intensities.

C. We added the following sentences in lines 192–194:

There are differences between runs 1 and 6 and between runs 7 and 8, although the oxidation conditions are similar; these will be due to differences in initial reactant concentrations or uncertainties resulting from sensitivity variations.

**Comment 4**

*Authors present the Csat distributions as sum of particle and gas phase (Figure 4) where particle phase is based on measured composition and gas phase is calculated assuming gas-particle equilibrium. Based on Figure 3, O/C is continuing to change at the end of the experiment suggesting that the system is not in equilibrium. Based on text on page 6 it's not clear if this was considered in the analysis when estimating abundance of gas phase compounds.*

**Reply**

The equilibration time scale of gas–particle partitioning in this study is less than 1 h, as discussed in the reply to comment 3 made by the other reviewer. The O/C ratio increased from 0.340 to 0.355 for 1 h (i.e., 4% h^-1) during a filter sampling period in run 8. In other words, the variation of O/C is small ($\leq$ 4%) during the gas–particle equilibration; therefore, we can assume gas–particle equilibrium even though O/C is continuing to change at the end of the experiment.

**Comment 5**

*The volatility distributions in Fig. 4 a-b have different shape compared to Fig. 4 c. Do authors have any ideas on what causes this?*

**Reply**

In the revised manuscript, we discussed the volatility distribution of LC/MS further, and we calculated additional results employing a third saturation concentration parameterization method (Fig. 4c). We also check the sensitivity of transmission efficiency (Fig. 4d). We compared the results in Figs. 4a, 4b, 4c, and 4d with the results of TD-AMS (Fig. 4e), and discussed potential reasons for the difference in the shape of the volatility distribution. We added the following explanations in lines 314–318 of the manuscript:

The volatility distributions in Figures 4a, 4b, 4c, and 4d (LC/MS) have different shapes than Figure 4e (TD-AMS). The shapes of the volatility distributions obtained from LC/MS analysis may be affected by uncertainties in saturation concentration and sensitivity parameterizations as well as the existence of undetected molecules. The shape of the volatility distributions obtained from the TD-AMS may be influenced by heat-induced reactions.

**Comment 6**

*Heat induced evaporation (Figure S3 and P7 L248-260): The variation of MFR at T=50C seem rather large for cases with OH scavenger (black markers). Did the authors look into what could cause this variation? Authors state that "Although the effects of OH scavengers, photochemical aging, and relative humidity on particle volatility were studied, the thermograms showed that all SOA results were similar to each other, within experimental uncertainties." Can the variation between the black markers be something else than experimental uncertainty? In that case there could be some effect of OH scavenger, aging and RH hidden in the data points.*

**Reply**

A probable reason of data variations is the error of thermodenuder temperature. If this is the case, all TD-AMS data have errors of similar level; therefore, we cannot discuss further on the effect of OH scavenger, aging, and RH. We added the following sentences in lines 300–302:

The variations between black symbols at 50 ° C are greater than those at higher temperatures. A major reason for such variations may be the difference between the sensor temperature and the TD effective temperature.

**Comment 7**

*"The present results indicate that gas/particle partitioning was virtually irreversible even though the VFR continued to decrease after 3 h." (P7, 282) This is unclear. What do the authors mean by irreversibility here?*

**Reply**

What we describe here is that SOA mass concentration after dilution does not agree with that predicted by the yield curve measured in SOA growth experiments. This sentence was also highlighted by the other reviewer. We have decided to remove this sentence in the new manuscript. Please see also our reply to comment 3 made by the other reviewer.

**Comment 8**

*"To the best of our knowledge, this study is the first to arrive at the results described above." (P8, L291) Please be more specific here. Not all the results described in the manuscript are as new as this suggests. Highly oxidized and/or long carbon chain compounds and slow evaporation are detected in earlier studies also, e.g. in the references of the current manuscript.*

We revised sentences in lines 359–362 of the manuscript as follows:

To the best of our knowledge, this study is the first to analyse HOMs by column separation and to compare the product volatility distribution determined by chemical analysis with those determined by particle evaporation measurements. The HOM detection by column separation is a valuable contribution to the current research because this technique could potentially be applied to molecular identification.

**Comment 9**

*Please consider revising the title "Lower than expected volatility of secondary organic aerosols formed during -pinene ozonolysis". The evaporation of particles was indeed slower than expected based on the yield experiments. However, considering Vaden et al. (2011, ACP) and other studies since then (which authors reference in the manuscript), I don't think "lower than expected" is anymore valid. With this I do not by any means suggest that the current study would not be important. It is definitely a topic which is worth of extensive research. However, the current title suggests that the authors find even lower volatility than Vaden et al. and other studies have found, which doesn't seem to be the case.*

**Reply**

We agree with this comment and replaced the title with the following new title:

Studying volatility from composition, dilution, and heating measurements of secondary organic aerosols formed during α-pinene ozonolysis

**Comment 10**

*Figure S2 and Table S2. These include only a small number of compounds and different compounds for different methods. It is not clear why these specific compounds were chosen for these comparisons.*

**Reply**

The results calculated by EVAPORATION are employed for the fitting by a 1D function. Chemical structure information is necessary for predictions of saturation concentration, thus α-pinene oxidation products of which chemical structures have been previously suggested, including semi-volatile compounds, highly oxygenated molecules, and dimers, were selected for EVAPORATION calculations. We assumed that the results of parameterization were applicable to all products detected by LC/MS. EVAPORATION data were validated against the results of a different calculation method, the SPARC online calculator (Hilal et al., 2003), and experimental results of PTR-MS. The SPARC data were calculated for α-pinene oxidation products which have already been identified in previous studies. All compounds selected in the SPARC calculations are included in the group of compounds selected for the EVAPORATION calculations. The PTR-MS results include only data for compounds that are detected in the region $151 \leq m/z \leq 229$; gas compounds are only detected by PTR-MS in the region $m/z \leq 229$; and to avoid interference from fragment ions, we only used results of $m/z \geq 151$. The PTR-MS data include results of both identified and unidentified products. We added the following sentences in lines 228–233 of the manuscript and added a brief explanation in the caption of Figure S2 of the supplementary material:

Chemical structure information is necessary for predictions of saturation concentration, thus α-pinene oxidation products for which chemical structures have been previously suggested, including semi-volatile compounds, highly oxygenated molecules, and dimers, were selected for EVAPORATION calculations. We assumed that the results of parameterization were applicable to all products detected by LC/MS. EVAPORATION data were validated against the results of a different calculation method, the SPARC online calculator (Hilal et al., 2003), and experimental results of PTR-MS.

In addition, we added the following sentences in lines 112–115 of the manuscript:

The ion signals of $m/z$ 151–229 were only used for the evaluations of saturation concentration. The signals of $m/z < 151$ were not used because there would be interference from fragment ions. The signals of $m/z > 229$ were not detected due to the low sensitivity of the quadruple mass spectrometer.

**Technical comment 1**

*P1, L22: "Atmospheric fine aerosols are believed to negatively affect climate (IPCC, 2013): : :" Is this a typo? What do authors mean by 'negatively affect climate'?*

**Reply**

We removed "negatively" from the sentence.

**Technical comment 2**

*P1, L39: "Wilson et al., 2014" should probably be 2015.*

**Reply**

You are correct: we revised it.

**Technical comment 3**

*P2, L41-43: ": : :however, later research showed that the evaporation process is strongly influenced by the particle phase state, and suggested that dilution results cannot be simply interpreted by gas/particle partitioning." Please add reference to this statement.*

**Reply**

We have revised this paragraph in response to specific comment 1 made by the other reviewer. We added references at appropriate places in the revised text.

**Technical comment 4**

*P5, L179: "The highest relative abundance was observed in the photochemical aging experiment,: : :" Please specify which compounds you refer to.*

**Reply**

We specified the compounds we refer to in the revised manuscript.

**Technical comment 5**

*P5, L191: ": : :that such dimerization will occur prior to the formation of the latter."Please clarify this sentence.*

**Reply**

To clarify this, we revised the sentence in lines 208−209 of the manuscript as follows:

The results suggest that dimers will be formed from less-oxygenated monomers instead of HOMs, and that dimerization will occur prior to the formation of monomer HOMs.

**Technical comment 6**

*P7, L271: "Although evaporation is assumed to occur instantaneously in VBS models,: : :" Did you mean equilibrium partitioning models? VBS is often used also for representing dynamics of condensation/evaporation.*

**Reply**

Yes, we do. We replaced VBS models with "equilibrium partitioning models" here.

**Technical comment 7**

*P8, L297: ": : :and photochemical aging in gas phase would be slower than that expected in the model." Please clarify this sentence. It is not clear how this conclusion was reached.*

We totally revised this paragraph according to specific comment 12 by the other reviewer. This sentence was removed in the revised manuscript.

**Technical comment 8**

*Figure S2. At the start of figure caption, please change "MW" to "Molecular weight (MW)" for clarity.*

**Reply**

We have revised it accordingly.

**References**

Ehn, M., Thornton, J. A., Kleist, E., Sipilä, M., Junninen, H., Pullinen, I., Springer, M., Rubach, F., Tillmann, R., Lee, B., Lopez-Hilfiker, F., Andres, S., Acir, I.-H., Rissanen, M., Jokinen, T., Schobesberger, S., Kangasluoma, J., Kontkanen, J., Nieminen, T., Kurtén, T., Nielsen, L. B., Jørgensen, S., Kjaergaard, H. G., Canagaratna, M., Dal Maso, M., Berndt, T., Petäjä, T., Wahner, A., Kerminen, V.-M., Kulmala, M., Worsnop, D. R., Wildt, J., and Mentel, T. F.: A large source of low-volatility secondary organic aerosol, Nature, 506, 476–479, doi:10.1038/nature13032, 2014.

Faulhaber, A. E., Thomas, B. M., Jimenez, J. L., Jayne, J. T., Worsnop, D. R., and Ziemann, P. J.: Characterization of a thermodenuder-particle beam mass spectrometer system for the study of organic aerosol volatility and composition, Atmos. Meas. Tech., 2, 15–31, doi:10.5194/amt-2-15-2009, 2009.

Fujitani, Y., Saitoh, K., Fushimi, A., Takahashi, K., Hasegawa, S., Tanabe, K., Kobayashi, S., Furuyama, A., Hirano, S., and Takami, A.: Effect of isothermal dilution on emission factors of organic carbon and n-alkanes in the particle and gas phases of diesel exhaust, Atmos. Environ., 59, 389–397, doi:10.1016/j.atmosenv.2012.06.010, 2012.

Pankow, J.F.: An absorption model of gas/particle partitioning of organic compounds in the atmosphere, Atmos. Environ. 28, 185–188, doi:10.1016/1352-2310(94)90093-0, 1994.

Sato, K., Hatakeyama, S., and Imamura, T.: Secondary organic aerosol formation during the photooxidation of toluene: NOx dependence of chemical composition, J. Phys. Chem. A, 111, 9796–9808, doi:10.1021/jp071419f, 2007.

Sato, K., Jia, T., Tanabe, K., Morino, Y., Kajii, Y., and Imamura, T.: Terpenylic acid and nine-carbon multifunctional compounds formed during the aging of b-pinene ozonolysis secondary organic aerosol, Atmos. Environ., 130, 127–135, doi:10.1016/j.atmosenv.2015.08.047, 2016.

Shiraiwa, M., Berkemeier, T., Schilling-Fahnestock, K. A., Seinfeld, J. H., and Pöschl, U.: Molecular corridors and kinetic regimes in the multiphase chemical evolution of secondary organic aerosol, Atmos. Chem. Phys., 14, 8323–8341, doi:10.5194/acp-14-8323-2014, 2014

Vaden, T. D., Imre, D., Beránek, J., Shrivastava, M., and Zelenyuk, A.: Evaporation kinetics and phase of laboratory and ambient secondary organic aerosol, Proc. Natl. Acad. Sci. USA, 108, 2190–2195, doi:10.1073/pnas.1013391108, 2011.

Wilson, J., Imre, D., Beránek, Shrivastava, and Zelenyuk, A.: Evaporation kinetics of laboratory-generated secondary organic aerosols at elevated relative humidity, Environ. Sci. Technol., 49, 243–249, doi:10.1021/es505331d, 2015.

Table R1. A summary of the amounts of consumed reactants, gaseous products, and SOAs.

|  | Run 5 | Run 6 | Run 7 | Run 8 |
|---|---|---|---|---|
| $\Delta[\alpha\text{-pinene}]$ (ppmv) | 0.10 | 0.31 | 0.30 | 0.14 |
| $\Delta[O_3]$ (ppmv) | 0.08 | 0.27 | 0.19 | 0.15 |
| $\Delta[M]_c$ (mg m$^{-3}$) | 0.72 | 2.3 | 2.0 | 1.1 |
| [SOA] (mg m$^{-3}$) | 0.22 | 0.96 | 0.86 | 0.30 |
| $\Delta[M]_c - [SOA]$ (mg m$^{-3}$) | 0.50 | 1.3 | 1.1 | 0.8 |
| $\Delta[Gas]_{PTR\text{-}MS}$ (mg m$^{-3}$) | 0.33 | 1.0 | 1.2 | 0.48 |
| $\Delta[Aerosol]_{PTR\text{-}MS}$ (mg m$^{-3}$) | a | 0.30 | 0.25 | a |

a: The filter sample was not analyzed by the PTR-MS.

---

## Author Comment (AC2) · 22 Jan 2018

**Reply to Prof. Grieshop**

Kei Sato, Yuji Fujitani, Satoshi Inomata, Yu Morino, Kiyoshi Tanabe, Sathiyamurthi Ramasamy, Toshihide Hikida, Akio Shimono, Akinori Takami1, Akihiro Fushimi, Yoshinori Kondo, Takashi Imamura, Hiroshi Tanimoto, Seiji Sugata

**Note and general comments**

*Note: I am a referee for the paper but am posting my review with my name as much of my response is informed by a highly related (and complementary) paper that a former student and I published on a similar topic (Saha and Grieshop 2016). It appears that the authors of this paper may not be aware of our paper, and I wished to be transparent so thought it most appropriate to post this review under my name.*

*This paper presents results from a series of chamber experiments in which a-pinene was oxidized with ozone and OH and the resulting aerosol measured in-situ and via measurements of particulate matter. A-pinene ozonolysis is a well-studied system, but a number of open questions remain, with the product distribution (typically quantified in terms of volatility) and phase-partitioning kinetics being among two of the most pressing ones. This paper adds to the literature discussing this system by including cutting-edge chemical analyses of particles formed from both ozonolysis and particles thus formed and then perturbed via OH oxidation and additional dilution. A number of different experimental arrangements – e.g. use of thermodenuder (TD), analysis of PTR-MS measurements of gas-phase and vapors driven off a filter, ESI-MS analysis of filter samples and multi-chamber dilution studies – provide valuable data sets by which the partitioning and chemical composition of this SOA can be probed. This is a system and a topic of great interest to ACP readers, and as such I believe this is highly appropriate venue for this analysis. This is an interesting and creative set of experiments resulting in data collected with cutting edge techniques. However, while the data appear to be carefully collected, I find some of the interpretation of these data could be improved and that some of the resulting conclusions are insufficiently supported by the analysis. Therefore, I suggest that this paper be substantially revised before publication is considered.*

**Reply**

Thank you for your valuable comments. We very much appreciate the time and effort you spent reviewing our paper and welcome the opportunity to improve its clarity and accuracy. We apologize that we did not refer to your recent work (Saha and Grieshop, 2016) in the original manuscript; we have rectified this oversight and now cite your results.

**Comment 1**

*The title and tone of discussion don't give proper weight to previous work that has shown that the partitioning of α-pinene SOA is not well-represented by 'traditional' yield experiments. As the other reviewer suggested, this is not a 'new' insight, and so more effort should be made to put the present results into the proper context, including that provided by our paper and others with more detailed chemical analyses that well preceded ours.*

**Reply**

We agree with the comment. Taking into account these comments and comment 9 by the other reviewer, we replaced the title with the following new one:

Studying volatility from composition, dilution, and heating measurements of secondary organic aerosols formed during α-pinene ozonolysis

**Comment 2**

*Inter-experiment variation is an important theme that should be explored. There is relatively little comparison between sets of experiments conducted at similar conditions to establish repeatability of efforts (which seem to be good for TD results, based on Runs 2, 3 and 4 in Fig. 4c) and also to more systematically explore the influence of different experimental conditions (e.g. OH scavenger, RH) on the measurements. One complication, especially to interpreting TD results is that OA concentration varies quite substantially (Table 1). Our paper showed that smog chamber OA concentration has a systematic effect on its observed evaporation. These data are collected at a range of concentrations, though a number of other parameters were simultaneously changed and so it's difficult to say whether the effect of individual changes can be extracted. In addition to inter-experimental variability, I'm also curious about possible explanations for the quite substantial variation in volatility distributions from different methods (Fig. 4 panels) – are there any cases in which experiments show better or worse agreement? What might explain these differences?*

**Reply**

Responding to this comment and comment 10 by you, we cited your paper and added the following discussion on the dependence of TD data on mass concentration. The following text is added in lines 296−302:

Saha and Grieshop (2016) reported that SOA volatility increases with increasing mass concentration in the range of 5–445 μg m$^{-3}$. Although we also obtained TD-AMS data between 964 and 2,400 μg m$^{-3}$ in runs 1, 2, 3, 4, and 6 (black symbols), our results showed no clear trend with mass concentration. We

surmise that the observed dispersion is probably caused by either the large variation in the present MFR data or differences in the mass concentration ranges between the current and previous studies. The variations between black symbols at 50 °C are greater than those at higher temperatures. A major reason for such variations may be the difference between the sensor temperature and the TD effective temperature.

Furthermore, we added discussion about experimental uncertainties in volatility distributions obtained by different methods. In the revised manuscript, we discussed the volatility distribution of LC/MS further, and we calculated additional results employing a third saturation concentration parameterization method (Fig. 4c). We also check the sensitivity of transmission efficiency (Fig. 4d). we compared the results in Figs. 4a, 4b, 4c, and 4d with the results of the TD-AMS analysis (Fig. 4e), and discussed potential reasons for differences in shape of the volatility distributions. Following this comment and comment 5 by the other reviewer, the following text has been added in lines 315–318:

The volatility distributions in Figures 4a, 4b, 4c, and 4d (LC/MS) have different shapes than Figure 4e (TD-AMS). The shapes of the volatility distributions obtained from LC/MS analysis may be affected by uncertainties in saturation concentration and sensitivity parameterizations as well as the existence of undetected molecules. The shape of the volatility distributions obtained from the TD-AMS may be influenced by heat-induced reactions.

**Comment 3**

*I have some major concerns about the collection and analysis of the room temperature evaporation data. In considering these results, I'd ask the authors to take a careful look at our paper (Saha and Grieshop 2016) and also work by Saleh et al. (Saleh et al. 2011, 2013), especially their chamber experiments considering time scales of equilibration. An important point is that equilibration rates cannot be readily separated from volatility in this type of experiment, and so the authors should not compare the assumed equilibrium of particles calculated based on yield experiment data (horizontal lines in Fig. 5) with dynamic equilibration data unless the consistency is somehow proven (e.g. the actual equilibrium state of the system is known). This disconnect especially stands out here because the authors have presented volatility distributions showing much lower volatility, so it is strange to compare 'equilibrium' dictated by another experimental approach (yield experiments) when other parts of your paper are essentially arguing that these are wrong. A better way to compare equilibrium properties is by comparing volatility distributions. Normalized time scales (see Saleh et al. papers referenced above) can be used to compare equilibration time scales for such experiments. As it is, this figure suggests that equilibration time scales for the low and high RH experiments are very different, but this difference may be due to differences in volatility or other experimental conditions. If this is the case, the volatility distributions in Fig. 4 may be*

*able explain this difference. As both our paper and (Saleh et al. 2013) point out, if you assume the wrong volatility you can easily conflate 'slow' evaporation due to a kinetic limitation (as you suggest), when it's really just that (as your chemical analyses suggest) the aerosol is much less volatile than yield data suggest. This is an assumption that we show may be at play in the 'kinetic limitation' ascribed to slow evaporation by Vaden et al. (2011). In addition, since different chambers (Teflon-coated steel vs. FEP membrane) were used for the different experiments, it's impossible to say what influence different wall characteristics may have of on the observed evaporation. As it is, there is far too little evidence to support the statements on Line 279-284 about 'irreversible' partitioning and the RH-dependence of evaporation and viscosity. There is good evidence elsewhere that RH dependencies may be important, but I don't find what is presented here convincing. For example, the data don't support the statement (L282-283) that that 'gas/particle partitioning was virtually irreversible'. This level of dilution is insufficient to probe the irreversibility of partitioning for compounds with C* less than 10ˆ1 or 10ˆ2 ug/m3, which comprise the majority of the material in your distribution (Fig. 4) but not the Lane et al. (2008) distribution used to calculate equilibrium partitioning.*

**Reply**

Following the comment, we carefully read papers the suggested papers (Saha and Grieshop (2016) and Saleh et al. (2011; 2013)). Revisions are made to descriptions in the introduction as described in the reply to specific comment 1. We removed straight lines drawn in Fig. 5 because we agree with your suggestions. We determined equilibration scaling time to be 24–46 min for the present dilution data. Scaling results are shown in Figure R1. We revised the last paragraph of section 3.4 (lines 334–346 of the manuscript) as follows to discuss scaling factors measured in present study:

The equilibration scaling time, defined by Saleh et al. (2011; 2013), was used to characterize slow SOA evaporation. The equilibration scaling time was determined to be 24–38 min and 33–46 min for α-pinene SOA particles generated in dry and humid experiments, respectively. The results are not compared between dry and humid experiments here because the chamber systems differ between these experiments. The current results of a 24–46 min equilibration time scale imply that the gas–particle equilibrium approximation could be applied for evaporation of α-pinene SOAs under atmospheric conditions, as suggested by previous workers (Saleh et al., 2013; Saha and Grieshop, 2016). The theoretical equilibration scaling time was also evaluated using eq. 3 in Saleh et al. (2013), where the accommodation coefficient was set to a recommended value of 0.1 for α-pinene SOAs. The theoretical equilibration scaling time was determined to be 24–41 min, which was similar to the experimental results: 24–46 min. These results suggest that kinetic inhibition may be a possible reason of slow evaporation. The present results based on SOA particles contrast with previous results based on semi-volatile diesel particles, which evaporate instantaneously after dilution (Fujitani et al., 2012). Kinetic inhibition during

dilution-induced SOA evaporation may occur because SOA particles include low or extremely low volatility organic compounds.

Consequently, we also revised the abstract. The text in lines 18–22 of the manuscript has been revised as follows:

The volume fraction remaining of SOAs decreased with time and the equilibration time scale was determined to be 24 – 46 min for SOA evaporation, suggesting that kinetic inhibition may be a possible reason for slow evaporation. The kinetic inhibition may occur because SOA particles contain compounds that are less volatile than those of semi-volatile products.

We determine volatility distributions from dilution data in the region 1–1000 μg m$^{-3}$ because the dilution data obtained in present study are only available in a region C* >1 μg m$^{-3}$. These results are added to Figure 4 of the revised manuscript and Table S3 of the supplementary material. We discuss these results in lines 347–354 as follows:

The volatility distributions determined from dilution data are compared with those determined from SOA yield curves (Lane et al., 2008) as shown in Figure 4f. We determined the volatility distributions from dilution data assuming gas–particle equilibrium using the VFR data measured 3 h after the dilution. The volatility distributions were only calculated in the range 1-1000 μg m$^{-3}$ because dilution data are only available for $C^{*}$ > several μg m$^{-3}$. The average $\log_{10} C^{*}$ values determined from dilution measurements (1.00–1.60) are lower than that determined from SOA yield curves (2.26); these results of dilution experiments might be consistent with the results of LC/MS and TD-AMS. However, note that the volatility determined from dilution experiments will be underestimated due kinetic inhibition.

Furthermore, we removed discussion of reversibility. We only discussed slow evaporation using scaling factors as described above.

**Reply**

**Specific comment 1**
*L42-43 – Discussion of dilution results is limited, and the body of literature lumped as 'later research' should be further discussed. This is a logical place to discuss Saleh et al. (2013).*

**Reply**
We added the citations of Saha and Grieshop (2016) and Saleh et al. (2013) and revised sentences in lines

40–47 as follows:

Yet another technique used to study volatility distribution is dilution-induced evaporation. Grieshop et al. (2007) diluted SOA particles in a reactor and studied the reversibility of gas–particle partitioning. Later workers (Vaden et al., 2011; Saleh et al.; 2013; Wilson et al., 2015; Yli-Juuti et al., 2017) diluted SOA particles instantaneously in an external chamber. Saleh et al. (2011; 2013) defined the equilibration time scale of SOA evaporation, and reported that the equilibration time scale is several minutes to several tens of minutes for α-pinene SOA particles (Saleh et al., 2013). Slow evaporation could be due to the presence of low-volatility materials in SOAs, kinetic inhibition, or some combined effect (Saha and Grieshop, 2016). Therefore, data from dilution-induced evaporation measurements would be determined not only by product volatility but also by the particle phase.

**Specific comment 2**
*L90 – It would be helpful to compare density numbers determined here with those from other studies. In general, more comparison of results with the wealth of other studies of this model aerosol system would help to place this study in its proper context relative to the literature.*

**Reply**
We added the following text in lines 95–96 of the manuscript:

The density determined in this study is close to literature values: $1.32 \pm 0.10$ g cm$^{-3}$ (Ng et al. 2007) and $1.24 \pm 0.03$ (Malloy et al., 2009).

**Specific comment 3**
*L97-98 - Discussion of the use of pinonic acid as a reference for the thermodenuder (TD) measurements should be discussed, along with more generally the approach used for interpreting TD measurements to yield Fig 4c. Is the T_50 method of Faulhaber et al. (2009) used with only a single 'calibration' compound (pinonic acid) to develop the calibration? Or were more calibration compounds used? Was the calibration from Faulhaber used directly? If so, do the results for your single calibration compound match well with that from their calibration? It is not clear if your TD has exactly the same residence time as the one used there, which would be an important pre-requisite for applying this calibration.*

**Reply**
We used the results of T-50 calibration from Faulhaber et al. (2009) directly because the present TD has a residence time close to that used for their TD. We confirm their calibration results explain the results of the TD-AMS measurements of pinonic acid particles well. A T-50 of pinonic acid particles was

determined to be 35.4 deg C; and the $\log_{10} C^*$ value was calculated to be $1.7 \pm 1.0$ by a method of Faulhaber et al. (2009). The result of $\log_{10} C^*$ determined by this method was close to that determined for pinonic acid by SPARC (2.25). We added the following text in lines 308−311 of the manuscript:

We used the results of the calibration curve from Faulhaber et al. (2009) directly because the TD in our study has a residence time close to that used for their TD. We confirmed that their calibration results are consistent with our results for pinonic acid particles.

**Specific comment 4**

*Line 102-104 – I would be curious to see how wall losses of vapors might affect the determination of the distribution of gas-phase constituents in the chamber at the end of the experiment and how this, and also sampling/storage conditions might affect what was sampled on the chamber walls. Also, during the desorption of the filters, was the PTR-MS signal observed to return to background? In both your and our work, we find materials that remain in the particle phase (not on filters) in a TD to up to 120 C (393 K) with substantial residence times, so I'd be curious to see how PTR-MS-identified products evolve during desorption and if everything was actually desorbed from the filters. In general, no PTR-MS data are shown apart from the points shown in Fig. S2 in the supplement. Is there any reason these data are not further discussed?*

**Reply**

From PTR-MS data, we determined the saturation concentrations only for the compounds that are detected between m/z 151 and 229. The PTR-MS data were not used to determine the volatility distribution of gaseous products in this study. When we measure vapors in the chamber, the ion signals of m/z 151−229 decreased slowly due to wall loss after α-pinene is consumed by the ozonolysis; however, the lifetime of the detected compounds were longer than the gas-particle equilibration time scale (that is determined to be <1 h in this study). This suggests that we can apply the assumption of gas-particle equilibrium for the detected compounds. We estimated the maximum fraction of the total vapor wall loss to the total vapor mass. As per the reply to the comment 1 made by reviewer #1, we compared the amount of gaseous products detected by the PTR-MS with an expected value, which was calculated from a difference between the amounts of consumed reactants and formed SOAs. We estimated the loss of gaseous compounds to be up to 40 %.

The PTR-MS data in Figure S2 were estimated from the results of run 6. The thermogram of the filter analysis and the temporal variations of ion signals at m/z 185 and m/z 187, which are attributed to pinonic acid and pinic acid, respectively, are shown in Figure R2. As shown in the figure, the PTR-MS signal was observed to return to background levels. The temperature dependence of vapor pressure of pinic acid over

a flat surface has been reported by Bilde and Pandis (2001). The vapor pressure of pinic acid at 368 K (95 ºC) was calculated to be 0.19 Pa (= $1.4 \times 10^{-3}$ Torr). From the relationship between the absorption equilibrium constant and the vapor pressure of compound (Odum et al., 1996), the $F_{i,OM}/A_i$ ratio was derived as 0.1 (Mo was set to 960 µg m$^{-3}$ (run 6) as a maximum particle level at 95 ºC; $MW_{om}$ was set to 200; and, $\zeta$ was set to 1). As Prof. Grieshop pointed out, not everything might be desorbed from the particles; however ≥90 % of pinic acid exists in the gas phase at 368 K (95 ºC).

We did not discuss results of PTR-MS in detail because the results will be presented in a separate paper. We have added the following description in lines 112-115 of the manuscript.

The ion signals of m/z 151–229 were only used for the evaluations of saturation concentration. The signals of $m/z < 151$ were not used because there would be interference from fragment ions. The signals of $m/z > 229$ were not detected due to the low sensitivity of the quadruple mass spectrometer.

**Specific comment 5**

*Line 109 and 152-153 – A brief description of the analytical method would be helpful. For example, some mention of the source/significance of sodium adducts would be helpful, especially for those (like me) unfamiliar with this quirk of ESI analysis.*

**Reply**

We added the following text in lines 168–170:

Species that do not generate stable positive ions through protonation were ionized by clustering with Na$^{+}$ cations that are naturally present in the solvent chemicals and glassware (Kruve et al., 2013; Zhang et al., 2017).

**Specific comment 6**

*Line 241 – Error of +/- 2 – it is not clear what units this error is reported in? Possibly a factor of 2?*

**Reply**

This sentence explains the error of $\log_{10} C^*$. We revised this sentence (lines 270–272 in the new manuscript) as follows:

The $C^*$ values predicted for dimers by the 1D function agreed with the SPARC results within an error of two orders of magnitude; in contrast, the 2D function predicted $C^*$ values five to six orders of magnitude higher than the SPARC results.

**Specific comment 7**

*Line 212-215 – While a line can be drawn through these data points, and shows a reasonably high R^2 value, there seems to be a discontinuity in this relation at a logC\* of around -3 or -4, with the clusters of data at either side of this showing a much shallower relationship between MW and logC\*. For example, if you exclude the lower volatility data, you would have a much different dependence. How does this affect your results? Is there any reason for this discontinuity, perhaps due to the oxidation state or other properties of the compounds in these two clusters?*

**Reply**

The carbon oxidation states will differ significantly between the products of m/z >=300 and <300 based on the present LC/MS results. We treated data sets in these two regions separately and fitted two different straight lines. These results are shown in Figure S3 and Table S2 in the revised supplementary material. We discussed this third fitting method as follows in lines 242–245 of the manuscript:

The third parameterization technique is the binary fit method. In Figure S2, the linear function overestimates molecular weights in the region $\log_{10} C^* = -3$ to 0. There are significant differences between the carbon oxidation states for the products with m/z ≥300 and <300, according to our current LC/MS results. We fitted these two data sets separately for better approximation as shown in Figure S3.

Furthermore, the volatility distributions determined using this binary fit are shown in Figure 4 of the revised manuscript. We discuss these results in lines 275–281 of the manuscript as follows:

The volatility distributions determined using the binary fitting method (Fig. 4c) have different shapes compared to those obtained using the 1D and 2D fits, suggesting that the shape of the volatility distribution obtained by LC/MS data depends on the saturation concentration parameterization. Table S2 shows that the binary fit provides a better approximation for MBTCA than the 1D and 2D fits although the binary fit provides worse approximations for dimers than the 1D fit. Table S3 compares the average $\log_{10} C^*$ values determined for volatility distributions obtained in this study. The average $\log_{10} C^*$ values determined for the binary fit (-2.19 to -2.76) were close to or lower than those determined for 1D fit (-2.71 to -0.83) and lower than those determined for 2D fit (-0.61 to 1.44).

**Specific comment 8**

*L221-223 – This may be a very normal assumption to make for a TOF-MS, but I would like to see it justified, either with a reference or calibration data. How sensitive are results (e.g. volatility distributions) to any uncertainty in this?*

**Reply**

For example, the TOF-MS transmission correction is discussed in Heinritzi et al. (2016). We cite this paper in the revised manuscript. We calculated volatility distributions without transmission correction and added these results to Figure 4 of the revised manuscript and Table S3 of the supplementary material. We discuss these results in lines 282–287 of the manuscript as follows:

As a sensitivity check of the transmission correction, we calculated volatility distributions obtained by a 1D fit without accounting for the transmission correction (Figure 4d). The average $\log_{10} C^*$ values determined without the transmission correction (-3.55 to -1.38) are close to or lower than those determined with the transmission correction (-2.71 to -0.83). The average $\log_{10} C^*$ values determined for all LC/MS data (-3.55 to 1.44) are lower than those determined for the yield curve (2.66), suggesting that α-pinene SOAs have a lower volatility than that expected from yield curve analysis.

**Specific comment 9**

*L242-244 – I don't quite understand this statement. How were 2D function results 'adjusted'? I'm assuming this is referring to the range of compounds fit by the 2D function? This statement and its implications should be clarified.*

**Reply**

We revised this sentence (lines 272–274 in the manuscript) as follows:

The accuracy of the results predicted by the 2D function was worse than those predicted by 1D function because the 2D function was fitted not only to the data of α-pinene oxidation products but also that of other organic compounds.

In addition we revised the sentences in lines 239–241 of the manuscript as follows:

The 2D function was fitted to EVAPORATION data for various organic compounds including not only $\alpha$ -pinene oxidation products, but also other organic compounds present in the atmosphere.

**Specific comment 10**

*Line 250-255 – The level of agreement should be better described (perhaps by comparing others' data to yours). Also, I note that there is quite a bit of scatter in observed evaporation, especially at 50 deg. C, where there is nearly a factor of 2 range in VFR (~0.4 to 0.7) for the alpha-pinene SOA systems. It is stated that data were similar 'to within experimental uncertainties', but these uncertainties are not stated*

*or discussed. As noted above, comparing results collected for experiments at the same conditions (Coa, scavenger, RH, etc.) can constrain inter-experiment variability and then be used to discuss whether any observed differences may be ascribed to these conditions or uncontrolled variability in the experiments.*

**Reply**

A probable reason of data variations is the error of thermodenuder temperature. We added the following sentences in lines 300–302:

The variations between black symbols at 50 °C are greater than those at higher temperatures. A major reason for such variations may be the difference between the sensor temperature and the TD effective temperature.

**Specific comment 11**

*Line 291 – Our study and various others that our paper discussed and cite, many of which were also cited here (e.g. (Ehn et al. 2014; Zhang et al. 2015)) have pointed towards the prevalence of very low volatility compounds in pinene SOA in contrast to existing yield-experiment-based parameterizations. It may be true that you are the first who arrived at the results via application for ESI-MS, but that is not how I read this sentence.*

**Reply**

I revised these sentences (lines 359–362 of the manuscript) as follows:

To the best of our knowledge, this study is the first to analyse HOMs by column separation and to compare the product volatility distribution determined by chemical analysis with those determined by particle evaporation measurements. The HOM detection by column separation is a valuable contribution to the current research because this technique could potentially be applied to molecular identification.

**Specific comment 12**

*Line 293-294 – Since a VBS model by itself doesn't assume an evaporation rate, it's hard to make this comparison. Also, as noted, volatility and evaporation kinetics are conflated in the comparison in this paper, so insufficient evidence is presented to make this claim. In chemical transport models, the assumption is that frequently made that equilibrium partitioning adequately describes what happens within time steps of on the order of a ~hour. Our paper argues that this is probably a fine assumption for alpha-pinene SOA produced in our lab under atmospheric conditions, though the assumption cannot be made in interpreting thermodenuder measurements.*

**Reply**

We revised this paragraph (lines 363−368 of the manuscript) as follows:

The first-generation products formed during α-pinene ozonolysis were found to include compounds less volatile than those predicted from SOA yield curves, and the equilibration time scale of dilution-induced evaporation was found to be several tens of minutes. These findings support recent results of SOA chemical composition studies and SOA evaporation studies. In the current VBS model, the product volatility distributions determined by SOA yield curves are employed. Further improvement of the atmospheric simulation model will be necessary to explain both laboratory and ambient SOA levels.

**References**

Bilde, M. and Pandis, S.N.: Evaporation rates and vapor pressures of individual aerosol species formed in the atmospheric oxidation of α- and β-Pinene. Environ. Sci. Technol. 35, 3344−3349, doi:10.1021/es001946b, 2001.

[revised manuscript text omitted]

**Figure R1: SR_eff calculated form present dilution data plotted as a function of t/τ; where SR_eff is defined in eq. 1 of Saleh et al. (2013); t is time; and τ is equilibration time scale. The curve represents a first-order approximation to the dynamic response. This curve underestimates and overestimates experimental data in a region t/τ = 0–1 and 1–5, respectively, which is likely due to higher-order dynamic responses of a complex aerosol system.**

[Figure]

**Figure R2: (a) Thermogram of the filter analysis. Temporal variations of ion signals at (b) m/z 185 and (c) m/z 187 during the filter analysis.**

---

## Author Response (AR2)

**Reply to comments**

Kei Sato, Yuji Fujitani, Satoshi Inomata, Yu Morino, Kiyoshi Tanabe, Sathiyamurthi Ramasamy, Toshihide Hikida, Akio Shimono, Akinori Takami, Akihiro Fushimi, Yoshinori Kondo, Takashi Imamura, Hiroshi Tanimoto, Seiji Sugata

**Anonymous Referee #1**

**General comments**

*The authors have addressed my comments and revised the manuscript mostly sufficiently. I can recommend publishing the manuscript after minor modifications according to below comments.*

**Reply**

Thank you for your valuable comments; we appreciate the time and effort you have put into reviewing our paper. I have revised the manuscript based upon your input, as described below.

**Specific comments**

**Comment 1**

*My original comment 3 C concerned the large differences between same oxidation conditions in figure 3. To address this comment the authors have included the following sentence in the manuscript (192-194): "There are differences between runs 1 and 6 and between runs 7 and 8, although the oxidation conditions are similar; these will be due to differences in initial reactant concentrations or uncertainties resulting from sensitivity variations." It is not clear what the "uncertainties resulting from sensitivity variations" means here. It is also concerning if the different reactant concentrations can cause such differences e.g. between the runs 7 and 8. In figure 3 it even seems that the run 8 resembles more run 9 than run 7. This added sentence needs clarification in the manuscript. This is an important point to explain properly as the reader will need to consider how reliable the figure 3 results are and if they include reliable information on the effect of oxidation state on the composition.*

**Reply**

"Uncertainties resulting from sensitivity variations" mean the variation of mass spectrometer sensitivity during LC analysis. The sensitivity would have an uncertainty of ~10% due to these variations; however, it cannot fully explain the difference between the results measured in runs 7 and 8 for the C17 and C18 dimers with carbon oxidation state lower than -0.6. The sentences in lines 192-194 of the previous manuscript were incorrect; thus, I removed these sentences. Instead, we added new appropriate explanations.

We added the following sentence in lines 206-209:

With regard to nine- and ten-carbon HOMs, the result of run 7 was close to that of run 8. These results suggest that the relative abundances of nine- and ten-carbon HOMs in SOAs will be independent of the initial α-pinene concentration.

Furthermore, we added the following sentences in lines 221-225:

As for the $C_{17}$ and $C_{18}$ dimers with carbon oxidation state less than -0.6, the results of run 9 was close to or slightly lower than that of run 8. These results suggest that the addition of OH radicals will barely affect or slightly reduce the relative abundances of these dimers in SOAs. The result of run 8 was higher than that of run 7, suggesting that the relative abundances of these $C_{17}$ and $C_{18}$ dimers in SOAs may increase with decreasing the initial α-pinene concentration.

**Comment 2**

*L334-226: Please explain here how equilibration time scale was determined. I assume fitting the eq. (2) from Saleh et al. (2013) to the dilution data but this is not clear to a reader if they are not familiar with that paper.*

**Reply**

Your assumption on equilibration time scale calculations is correct. I have added the following sentence in lines 350-351:

The equilibration scaling time was determined by fitting eq. 2 in Saleh et al. (2013) to the dilution data.

**Comment 3**

*L347-354: It is not clear how the volatility distribution was calculated from the dilution data. This needs to be explained. What does it mean that "dilution data are only available for $C^* >$ several μg m-3"? Also, the statement "these results of dilution experiments might be consistent with the results of LC/MS and TD-AMS" does not seem justified. First, if volatility distribution from dilution data is determined so that only $logC^* > 0$ is included, then there is little ground for comparing that to the figures 4 a-e. Second, if one does such comparison anyway, consistency would require that the relative amounts of bins with $logC^*$ above 0 would be same between the distributions. This doesn't seem to be the case and even the volatility distributions from dry and 40% RH dilution experiments do not show such consistence between them two.*

We added Figure S6 to show how this distribution was determined. The following sentences describing

the procedure for determination of this distribution were added in lines 367-370:

The normalized SOA yields obtained in dilution experiments are plotted as a function of mass concentration (Figure S6). The volatility distribution was determined by fitting eq. 2 in Grieshop et al. (2007) to the plotted data.

The sentence in lines 370-371 was revised as follows:

The volatility distributions were only calculated in the range 1-1000 μg m$^{-3}$ because dilution data are only available in this region.

We removed the sentence, "these results of dilution experiments might be consistent with the results of LC/MS and TD-AMS."

**Comment 4**

*L366: What does "current VBS model" mean here? Do you mean the current VBS parameterizations used in global models?*

**Reply**

No we don't. We revised the sentence in lines 384-386 as follows:

In the standard VBS approach, the product volatility distributions determined by SOA yield curves are employed. Further improvement of the atmospheric simulation model will be necessary to explain both laboratory and ambient SOA levels.

**Referee #2: Andrew Grieshop, agrieshop@ncsu.edu**

**General comments**

*Overall, I'm quite happy with the updates that the authors made to the manuscript and believe that this will make a valuable addition to the literature. I have a number of hanging questions/concerns related to the way in which my original comments were addressed, listed below. Some of this is simply clarification, but I do have some concerns about how inter-experiment variation is treated, especially with regards to the thermodenuder measurements. If a potential reason for variation, it should be explored or supported with available evidence, rather than being presented as a supposition without supporting evidence. I would like to see these comments addressed in a revised manuscript.*

**Reply**

Thank you for your valuable comments; we appreciate the time and effort you have put into reviewing

our paper. We have revised the manuscript based upon your comments. Descriptions on the thermodenuder measurements have been revised based upon your comments 1 and 2.

**Specific comments**

**Comment 1**

*Line 301-302: This is not a satisfying explanation, unless it is experimentally verified... Based on sensitivity implied by line in Fig. S4, there would need to be a variation of ~25 C across these experiments, which is a lot. The 40 degree pinonic acid data don't have nearly so much scatter. This issue needs to be discussed further. Was the residence time in the TD closely controlled? Were calibration experiments on TD temperature control or experiments with single component aerosols conducted?*

**Reply**

No experimental evidence is available for the original explanation; therefore, I removed the sentences of the original explanation.

The MFR plotted in Figure S4 was defined to be the fraction of the mass concentration measured at a target temperature to that measured at $25^{o}C$; thus, MFR determined for $25^{o}C$ is defined to be unity in this study (there is no uncertainty).

The number of TD measurements of pinonic acid particles was much smaller than that of TD measurements of SOA. Thus, we cannot simply compare the variation of pinonic acid particle data with that of SOA particle data.

We did not monitor the flow rate during TD measurements although we check it prior to every day TD measurement. Thus, the variation of residence time may be a potential reason of the variation of present TD data.

We conducted the measurements of pinonic acid particles as described in the manuscript. The present pinonic acid particle measurements provided a saturation concentration consistent with the literature data (please see the reply to specific comment 3 given by Prof. Grieshop on the first discussion paper).

We show two examples of TD-AMS data. Figure R1 shows time series of SOA concentration measured by TD-AMS in runs 1 and 2. The TD equipped with a bypass line. During each cycle of TD-AMS measurement, we used the bypass for first 9 min to obtain the reference data and then used the TD of a specific temperature for 15 min. During the measurements shown in Figure R1, the TD mode was programmed to be bypass (0-3 min), TD of 50°C (3-18 min), bypass (18-27 min), TD of 100°C (27-42 min), bypass (42-51 min), TD of 150°C (51-66 min), bypass (66-75 min), TD of 200°C (75-90 min), and bypass (90-99 min). MFR was determined by accounting for SOA wall loss. As shown in Figure R1, the

bypass data decreased with time. In standard cases, the SOA concentration decreased by 5–9% for each measurement cycle (24 min) due to chamber wall loss. On the other hand, irregular fast decay was observed during the measurement of 50°C in run 2; the bypass data decreased by 19% for 24 min. Similar irregular results were also observed in runs 3 and 4. Irregular decay may result from the variation of the residence time which affects wall loss in the bypass.

Consequently, the variation of residence time may be a potential reason of the variation of present MFR data. I added the following sentences in lines 310-313:

The variation of MFR could be caused by the variation of the residence time. Although the flow rate was not monitored during present TD-AMS measurements, the SOA concentration was monitored through the bypass as the reference. The bypass result decreased by 5–9% for each TD measurement cycle (24 min) due to the SOA loss on the chamber wall (Figure S5); howerver, it decreased by 17–19% for 24 min in irregular cases observed during runs 2, 3, and 4. These results suggest that there may be the variation of the residence time in these runs.

In order to explain the bypass of TD, I also added Figure S2 and the following sentences in lines 103-105:

The TD was equipped with a bypass line. During each cycle of TD-AMS measurement, we used the bypass for first 9 min to obtain the reference data and then used the TD for 15 min to obtain the data of a specific temperature.

**Comment 2**
*Line 309-310: The description of residence time as 'close to' that of Faulhaber is vague. Please specify the residence time of the TD used in experiments. Was it help constant across all experiments?*

**Reply**
We explicitly describe the residence time in the sentence in lines 321-323 :

We used the results of the calibration curve from Faulhaber et al. (2009) directly because the TD in our study has a residence time (~13 s at room temperature) close to that used for their TD (~15 s at room temperature).

In our study, the flow rate was about 1 L min$^{-1}$. However, we did not control flow rate; this may also be a potential reason of TD data variation as described in the reply to your specific comment 1.

**Comment 3**
*Line 343: 'slow' is a vague term, as is kinetic inhibition. There is some kinetic inhibition, but this is not*

*slow relative to what others have claimed as time scales (e.g. Vaden et al 2011) or slow with respect to a time step in most atmospheric models.*

I removed the term, "slow", and revised this sentence in lines 361-362 as follows:

These results suggest that kinetic inhibition may explain the equilibration time scale measured in this study.

I also made a similar revision in the abstract.

**Comment 4**

*Line 343-344: Comparison with Diesel emission evaporation is not appropriate for comparison evaporation rates, as the materials likely have very different volatility distributions, and the change in vapor concentration will be a function of the underlying volatility distribution and the relative change in concentration. Evaporation time scales or evaporation coefficients can/should be compared instead.*

**Reply**

I removed this sentence. Instead, we revised the following sentence in the introduction (lines 40-42) to mention volatility studies of diesel exhaust particles.

Yet another technique used to study volatility distribution is dilution-induced evaporation, which has been successfully applied to the volatility studies of diesel exhaust particles (Robinson et al., 2007; Fujitani et al., 2012).

**Comment 5**

*Line 345-346: Kinetic inhibition being linked to thermodynamic properties is not appropriate, unless a mechanism is discussed. This is conflating thermodynamics (volatility) with kinetics (inhibition).*

**Reply**

I have added the discussion of the mechanism in lines 357-360 as follows:

The mass accommodation coefficient subsumes all resistances to gas−particle partitioning other than gas phase diffusion; and a mass accommodation coefficient smaller than 1 would indicate that the condensed phase is highly viscous and exhibits substantial kinetic limitations.

**Comment 6**

*Line 349-354: Not very clear how this distribution was determined. How many data points? Why not compare to our volatility distribution in Saha and Grieshop (2016)? If you are comparing VFRs after 3*

*hours, this should be something like 6\*tau (considering your tau are ~30 min), so I'm not so sure kinetic inhibition would be an issue. Wall losses may be?*

We added Figure S6 to show how this distribution was determined. The following sentences describing the procedure for determination of this distribution were also added in lines 367-370.

The normalized SOA yields obtained in dilution experiments are plotted as a function of mass concentration (Figure S6). The volatility distributions were determined by fitting eq. 2 in Grieshop et al. (2007) to the plotted data.

We showed the data from Saha and Grieshop (2016) in Figure 4f. We added the following sentences in lines 332-333:

Figure 4f includes the volatility distribution determined from the previous TD study (Saha and Grieshop, 2016).This previous result is similar to present TD-AMS results.

We completely removed the sentence, "However, note that the volatility determined from dilution experiments will be underestimated due kinetic inhibition."

**Comment 7**
*Line 366: There is no one 'current VBS model', and it's not true that every model necessarily exclusively uses chamber yields (though that is definitely the standard approach).*

**Reply**
I revised this sentence in lines 384-386 as follows:

In the standard VBS approach, the product volatility distributions determined by SOA yield curves are employed. Further improvement of the atmospheric simulation model will be necessary to explain both laboratory and ambient SOA levels.

[Figure]

**Figure R1: Time series of SOA concentration measured by TD-AMS during the heating measurement in (a) run 1 and (b) run 2; the TD mode was programmed to be bypass (0-3 min), TD of 50°C (3-18 min), bypass (18-27 min), TD of 100°C (27-42 min), bypass (42-51 min), TD of 150°C (51-66 min), bypass ( 66-75 min), TD of 200°C (75-90 min), and bypass (90-99 min).**

---

## Author Response (AR3)

**Reply to editor comments on acp-2017-860-R2**

Kei Sato, Yuji Fujitani, Satoshi Inomata, Yu Morino, Kiyoshi Tanabe, Sathiyamurthi Ramasamy, Toshihide Hikida, Akio Shimono, Akinori Takami1, Akihiro Fushimi, Yoshinori Kondo, Takashi Imamura, Hiroshi Tanimoto, Seiji Sugata

**General comment:**

*The authors have largely adequately addressed this last round of reviewer comments, for which they are to be commended. However, there are two specific responses that are not satisfactorily addressed or appropriately worded.*

Thank you for reading the manuscript carefully and providing useful comments. We revised the manuscript following the comments.

**Specific comment 1:**

*The first is:*

*"The mass accommodation coefficient subsumes all resistances to gas−particle partitioning other than gas phase diffusion; and a mass accommodation coefficient smaller than 1 would indicate that the condensed phase is highly viscous and exhibits substantial kinetic limitations."*

*A mass accommodation coefficient smaller than unity by no means indicates a highly viscous condensed phase with substantial kinetic limitation. I would refer the authors to Kolb et al., ACP, doi:10.5194/acp-10-10561-2010, 2010 for a thorough discussion of the meaning of mass accommodation coefficient. It is perfectly possible for mass accommodation coefficient to be very much lower than unity without a particle being highly viscous - for example the mass accommodation coefficient of water vapour to a lubricating oil particle would be extremely low, but the particle would be quite inviscid. Any particle exhibiting immiscibility with the condensing vapour (or with a tendency to outgas or "salt-out" the component because of strong non-ideality) could behave the same whilst being quite fluid. I would agree with the original reviewer's request not to conflate thermodynamics and kinetic inhibition in the absence of a mechanism but find the proposed mechanism unconvincing. I'd invite the authors to tighten up the argument.*

**Reply:**

I agree with the comments. I discussed potential reasons suppressing the accommodation coefficient, and added the reference suggested in the comments. I toned down our suggestion from the results of mass accommodation coefficient. I revised the sentences in lines 357-365 as follows:

The theoretical equilibration scaling time was also evaluated using eq. 3 in Saleh et al. (2013), where the accommodation coefficient was set to a recommended value of 0.1 for α-pinene SOAs. The theoretical equilibration scaling time was determined to be 24–41 min, which was similar to the experimental results: 24–46 min. The mass accommodation coefficient subsumes all resistances to gas–particle partitioning other than gas phase diffusion, for example, surface accommodation, deviation from Maxwell-Boltzmann molecular velocity distribution near the particle surface, and diffusion limitations in the condensed phase (Kolb et al., 2010; Saleh et al., 2013). The mass accommodation coefficient was determined to be less than unity, suggesting that the existence of low-volatility materials in SOAs, kinetic inhibition, or some combined effect may explain the equilibration time scale measured in this study.

Furthermore, I revised the following sentences in the abstract in lines 18-23:

The volume fraction remaining of SOAs decreased with time and the equilibration time scale was determined to be 24 – 46 min for SOA evaporation. The experimental results of equilibration time scale could be explained when the mass accommodation coefficient is assumed to be 0.1, suggesting that the existence of low-volatility materials in SOAs, kinetic inhibition, or some combined effect may explain the equilibration time scale measured in this study.

**Specific comment 2:**
*The second difficulty with the response is the repeated statement that there is a "standard VBS approach". There are a multitude of VBS approaches and the authors should be specific. One way they could do this is to anchor the statement on a reference that distinguishes the approach to which they are referring from other "non-standard" VBS treatments (by stating "standard VBS approach (e.g. xyz et al., 2005)").*

**Reply:**
I added references for "the standard VBS approach" and also discussed non-standard treatments citing references. I revised the sentences in lines 385-389 as follows:

In the standard VBS approach, the product volatility distributions determined by SOA yield curves are employed and no reactions are assumed to occur in the particle phase (e.g., Robinson et al., 2007; Lane et al.; 2008). Currently, only a limited number of non-standard treatments are available; e.g., Trump and Donahue (2014) took into account oligomer formation in the particle phase, and Yli-juuti et al. (2017) employed the product volatility distribution determined from dilution data.

**References**
Kolb, C. E., Cox, R. A., Abbatt, J. P. D., Ammann, M., Davis, E. J., Donaldson, D. J., Garrett, B. C., George, C., Griffiths, P. T., Hanson, D. R., Kulmala, M., McFiggans, G., PöPiipinen, I., Rossi, M. J.,

Rudich, Y., Wanger, P. E., Winkler, P. M., Worsnop, D. R., and O'Dowd, C. D.: An overview of current issues in the uptake of atmospheric trace gases by aerosols and cloud, Atmos. Chem. Phys., 10, 10561–10605, doi:10.5194/acp-10-10561-2010, 2010.

Lane, T. E., Donahue, N. M., and Pandis, S. N.: Simulating secondary organic aerosol formation using the volatility basis-set approach in a chemical transport model, Atmos. Environ., 42, 7439–7451, doi:10.1016/j.atmosenv.2008.06.026, 2008.

Robinson, A. L., Donahue, N. M., Shrivvastava, M. K., Weitkamp, E. A., Sage, A. M., Grieshop, A. P., Lane, T. E., Pierce, J. R., and Pandis, S. N.: Rethinking organic aerosols: Semivolatile emissions and photochemical aging, Science, 315, 1259–1262, doi:10.1126/science.1133061 2007.

Trump, E. R. and Donahue, N. M.: Oligomer formation within secondary organic aerosols: equilibrium and dynamic considerations, Atmos. Chem. Phys., 14, 3691–3701, doi:10.5194/acp-14-3691-2014, 2014.